



# Capturing Solid Earth and Ice Sheet Interactions: Insights from Reinforced Ridges in Thwaites Glacier

Luc Houriez[1,2,*], Eric Larour[2], Lambert Caron[2], Nicole-Jeanne Schlegel[3], Surendra Adhikari[2], Erik Ivins[2], Tyler Pelle[4], Hélène Seroussi[5], Eric Darve[1], and Martin Fischer[6]

[1]Department of Mechanical Engineering, Stanford University; Stanford, CA, USA.
[2]Jet Propulsion Laboratory, California Institute of Technology; Pasadena, CA, USA.
[3]NOAA/OAR Geophysical Fluid Dynamics Laboratory; Princeton, NJ, USA.
[4]Scripps Institution of Oceanography, University of California, San Diego; San Diego, CA, USA.
[5]Thayer School of Engineering, Dartmouth College; Hanover, NH, USA.
[6]Department of Civil and Environmental Engineering, Stanford University; Stanford, CA, USA.
[*]Corresponding author. Email: houriezl@stanford.edu

**Abstract.** The projected evolution of marine ice sheets is greatly affected by Gravitation, Rotation, and Deformation (GRD) effects over century timescales. In the Amundsen Sea sector, GRD effects cause viscoelastic solid Earth uplift and near-field sea-level fall, reducing the ice sheet mass loss. Spatiotemporal resolutions are critical for computational feasibility and accurately capturing solid Earth and ice sheet interactions. However, the sensitivity of coupled ice sheet and GRD models to these resolutions is not fully understood. Here, we investigate the influence of: (i) the spatial resolution of the ice sheet model, (ii) the spatial resolution of the GRD response, and (iii) the coupling interval between the ice sheet and GRD models. We consider two model setups with distinct mesh structures, surface mass balance and basal melt parameterizations. Our findings underscore the importance of feedback mechanisms at kilometer scales and decadal to sub-decadal timescales. Resolving bedrock topography at 2 km instead of 1 km results in sea-level projection differences of 7.1% by 2100 and 18.8% by 2350. We examine the influence of GRD effects on bedrock ridges to explain the noted sensitivities. In our most conservative setup, we find that bedrock uplift extends buttressing by up to 30 years on ridges located 34 and 75 km upstream of Thwaites' current grounding line. This mechanism plays a key role in reducing Thwaites' sea-level contribution by up to 53.1% in 2350. These findings underscore the critical need to reduce uncertainties in bedrock topography.

## 1 Introduction

Accurate sea-level projections are paramount to risk mitigation efforts for coastal communities. On decadal to century timescales, one of the main sources of uncertainty lies in estimating the contribution of West Antarctica, and particularly Thwaites Glacier, to global mean sea-level (GMSL) change (Seroussi et al., 2023). Under high emission scenarios, Antarctica's contribution to GMSL could reach 28 cm by 2100 and 4.4 m by 2300, relative to 2015 (Seroussi et al., 2024). In the fastest melting region of the ice sheet, Thwaites Glacier has displayed signs of early collapse (Joughin et al., 2014).

Evidence of interactions between ice sheets, sea-level and solid Earth (Gomez et al., 2015) has been shown to significantly affect the evolution of marine-terminating ice sheets over millennial (Konrad et al., 2015; de Boer et al., 2014, 2017; Pollard





et al., 2017; Gomez et al., 2018), and more recently centennial (Kachuck et al., 2020; Book et al., 2022; Larour et al., 2019) timescales. In the vicinity of the grounding line –the region in which grounded ice becomes afloat, these interactions mainly comprise stabilizing negative feedbacks from sea-level fall and bedrock uplift (Gomez et al., 2010, 2012, 2015; Adhikari et al.,

2014). The negative feedback may be regionally enhanced by low mantle viscosity (Barletta et al., 2018; Whitehouse et al., 2019; Coulon et al., 2021). Our understanding of these feedback effects and the resulting predictions for West Antarctica's evolution relies on increasingly sophisticated models that combine ice flow with Gravitation, Rotation and Deformation (GRD) mechanisms (also known as Glacial Isostatic Adjustment –GIA) in a single consistent framework, hereafter referred to as coupled model.

To date, coupled model studies have focused primarily on the influence of mantle rheology due to the high uncertainty of this parameter and its role in controlling the strength of GRD effects (Book et al., 2022). However, the influence of a variety of other modeling choices in coupled models remains elusive. These include: (i) the spatial resolution of the ice sheet model, (ii) the spatial or spectral resolution of the GRD response, (iii) the coupling interval (equivalent to the time step of the GRD model), and (iv) the spatial resolution of lateral variations in the mantle structure (where applicable). These parameters condition com-

putational cost and are essential to accurately represent and update the bedrock throughout coupled model simulations. In previous works, the influence of these parameters has not been consistently isolated. In a study highlighting the need to capture elastic GRD response at 2 to 4 km resolutions, the spatial resolutions of the ice sheet model and GRD response were not distinguished (Larour et al., 2019). This contrasted with another study which recommends a 7.5 km resolution for three-dimensional (3D) GRD models to keep the error in bed deformation and sea-level patterns under 5% (Wan et al., 2022). Hence, while

we anticipate that optimal resolution values (i.e., beyond which models converge) may vary depending on approximations for mantle and ice physics, we note that some of the apparent inconsistencies between previous studies may be caused by failing to isolate the effects of different modeling choices.

Here, we investigate the distinct influences of: (i) the mesh resolution of the ice sheet model, (ii) the spatial resolution of the viscoelastic GRD response, and (iii) the coupling interval, on the grounding line and GMSL contribution of Thwaites Glacier.

We further explore the implications of GRD effects on key bedrock features in the basin to build understanding for the observed sensitivities.

## 2 Methods

All simulations are performed using the Ice-sheet and Sea-level System Model (Larour et al., 2012, ISSM) which permits the coupling of an ice sheet model and a GRD model within a single consistent framework. Ice sheet models have several major

components, including mesh structure, basal melt and surface mass balance (SMB) parameterizations, which may affect the sensitivities of coupled simulations to (i), (ii), and (iii). To account for such effects, we report our results for 2 widely different coupled model setups labeled SLIM and PLUS. These have the same GRD model but different ice sheet models representative of the spectrum of complexity of modern ice sheet models.





## 2.1 Ice sheet models

In SLIM (Steady smb, Linear ocean melt, Isotropic Mesh), the ice sheet model uses a linearly depth dependent ocean melt parametrization (Seroussi et al., 2014b) and a 2 km uniform mesh structure over the Amundsen Sea sector. Finally, a constant SMB forcing corresponding to the average between 1979 and 2010 from the Regional Climate Model RACMO2.1 is applied (Meijgaard et al., 2008). In PLUS (Picop ocean melt, Locally refined mesh, Unsteady Smb), the ice sheet model features the cavity ocean melt model PICOP (Pelle et al., 2019) which captures the buoyant plume behavior of sub ice shelf water. Ocean

temperature and salinity for PICOP are forced through 2300 using data from the Community Earth System Model 2 (CESM2) (Danabasoglu et al., 2020) for the Shared Socioeconomic Pathway SSP5-8.5. PLUS also incorporates CESM2 SMB forcings through 2300 for the SSP5-8.5 scenario. From 2300 to 2350, SMB, ocean temperature and salinity are held constant at their 2300 value. For PLUS, an anisotropic mesh structure refined to 1 km in key bedrock areas is used in Amundsen Sea sector. For both setups, we run simulations from 2000 to 2350 using the Shallow Shelf Approximation. The ice sheet model's time

step is set to 2 weeks to capture rapid changes in the grounding line which is allowed to migrate according to a floatation threshold defined by hydrostatic equilibrium. Ice rheology and basal friction are set to match initial velocities to observed velocities (Rignot et al., 2014). Although our study focuses on Thwaites Glacier and the Amundsen Sea sector, the model's mesh cover the entire globe. For icecaps outside of the Amundsen Sea sector, annual ice mass change through 2350 are linearly extrapolated from GRACE trends taken between 2003 and 2016 (Larour et al., 2017). Background GIA signal from the Last

Glacial Cycle (the response to ice and ocean loading between 122 kyr before present and the start of the industrial revolution) is also incorporated as an additional trend for the solid Earth and sea-level motion using the values provided by (Caron et al., 2018). In line with (Gregory et al., 2019), we use the terminology GRD for contemporary and future adjustments of the solid Earth and sea-level, which are not included in the background GIA.

## 2.2 GRD model

The ice sheet model provides variations of ice loading to the GRD model at a regular time interval (e.g., 1 year) called the coupling interval. From these, the GRD model computes the resulting bedrock and geoid changes. These fields are then updated in the ice sheet model before it resumes its computations. GRD effects account for rotational feedback, self-attraction and loading of the barystatic sea-level (Adhikari et al 2016). The Extended Burgers Material (EBM) (Ivins et al., 2022) rheology is used in mantle layers between the core–mantle and lithosphere–asthenosphere boundaries to compute viscoelastic Love

numbers and derive the corresponding GRD patterns. This is achieved via the new GRD capabilities of the ISSM (Adhikari et al., 2016; Larour et al., 2020, 2012; Farrell and Clark, 1976).

The layers of the solid Earth model, in line with existing literature on regional mantle properties (Barletta et al., 2018), are described in Table 1. Although our model features a complete Earth structure down to the planet center in accordance to ISSM's Love number solver, we anticipate that the GRD response to mass changes in the Amundsen Sea sector is most sensitive to

properties of the lithosphere and asthenosphere and comparatively less to the rest of the mantle (Barletta et al., 2018) given the spatial scale of loading considered. In comparison, signal originating from the ocean and ice loads in the far-field is most





**Table 1.** Solid Earth parametrization. Elastic and density parameters are taken from a volumetric average of the Preliminary Reference Earth Model (Dziewonski and Anderson, 1981, PREM) within each layer. The Earth model also includes a solid inner core and a fluid outer core layer with a density of 10750 kg.m$^3$. Transient rheology properties include the relaxation strength parameter $\Delta$, the power exponent $\alpha$, and the low & high cutoff periods, $\tau_L$ and $\tau_H$, delimiting the timescales in which the transient relaxation regime operates.

| Layer | Radius Interval (km) | Viscosity (Pa.s) | Density (kg.m$^3$) | Shear Modulus (Pa) | $\Delta$ | $\alpha$ | $\tau_H, \tau_L$ |
|---|---|---|---|---|---|---|---|
| Lithosphere | $6321 - 6371$ | $\infty$ | $3.054 \times 10^3$ | $1.6347 \times 10^{11}$ | N/A | N/A | N/A |
| Asthenosphere | $6171 - 6321$ | $3.16 \times 10^{18}$ | $3.370 \times 10^3$ | $0.4857 \times 10^{11}$ | 3 | 0.5 | 7 yr, 54 min |
| Upper Mantle | $5701 - 6171$ | $2 \times 10^{20}$ | $3.701 \times 10^3$ | $0.6686 \times 10^{11}$ | 3 | 0.5 | 7 yr, 54 min |
| Lower Mantle | $3480 - 5701$ | $2 \times 10^{22}$ | $4.904 \times 10^3$ | $0.9342 \times 10^{11}$ | 3 | 0.5 | 7 yr, 54 min |

sensitive to the deeper mantle (Caron and Ivins, 2020) and thus our viscosity profile therein more closely aligns with global GIA models. Transient rheology (Caron et al., 2017) properties are set identically for all mantle layers below the lithosphere and are chosen in accordance with estimations given by (Ivins et al., 2020, 2023; Lau and Faul, 2019) in which further description

of the influence of each of these parameters is provided.

### 2.3 Baseline resolutions for the SLIM and PLUS setups

For each setup, we consider a set of resolutions that will act as a baseline for our sensitiviy analysis. Baseline values for SLIM are set as follows: 1 year coupling interval, 2 km mesh resolution for the ice sheet model and 1 km resolution of the GRD response (corresponding to a maximum Love number degree of 40,000). For PLUS, the parameterization is identical, except

for the ice sheet model's anisotropic mesh which sequentially progresses upstream of the initial grounding line. It is set to 1 km in areas with critical bedrock features, coarsens to 1.5–2 km in the grounding line retreat zone and reaches 3.5 km in regions where moderate ice velocities (< 250 m.yr$^{-1}$) are simulated at the end of the simulation (see appendix A1 for details).

### 3 Results

We first establish the baseline run for both the SLIM and PLUS setups and determine the role played by major coupled

model components. We then expose in more detail the mechanisms by which GRD stabilization occurs. Finally, we assess the sensitivity of the grounding line retreat and GMSL contribution to the spatiotemporal resolutions tested.

### 3.1 The baseline Viscoelastic run

We create a baseline Viscoelastic run for both the SLIM and PLUS setups which serves as a reference for our sensitivity study. Here, we examine the role played by GRD effects, mesh structure, basal melt and surface mass balance parametrization

in each setup. To quantify viscoelastic GRD stabilization induced by the transient EBM rheology described in the Methods





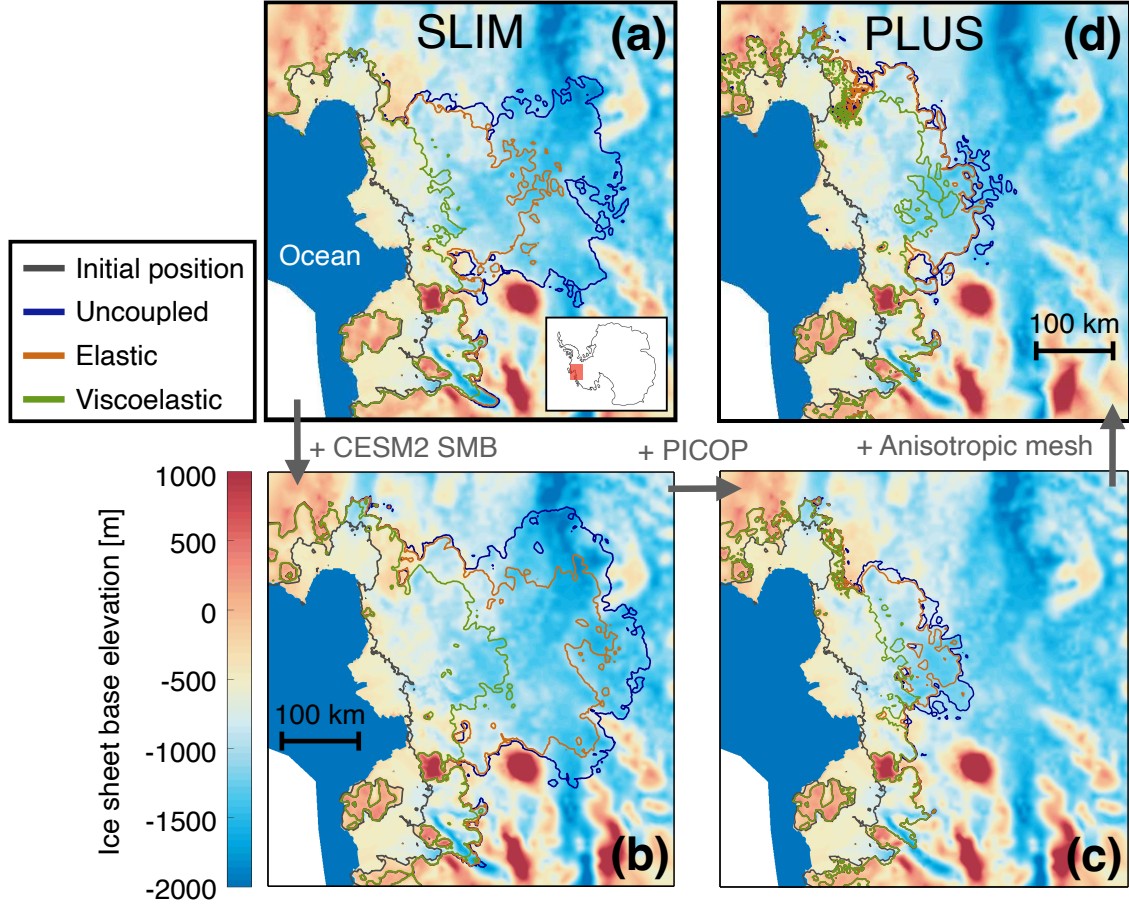

**Figure 1.** Grounding line projection in the Amundsen Sea sector at year 2350 for the SLIM (a) and PLUS (d) setups. Projections for intermediate setups are shown to illustrate the effects of incorporating CESM2 SMB (b), PICOP (c), and an anisotropic mesh. The grounding line positions for the Viscoelastic (green), Elastic (brown) and Uncoupled (blue) runs are overlaid on top of the bedrock topography. The initial grounding line position is represented in gray and the ice-free ocean is depicted in solid blue. The location of the basin in Antarctica is shown in the bottom right inset of (a).

section, we compare our baseline Viscoelastic run to two additional runs: the Uncoupled run in which no GRD effects are considered, and the Elastic run in which a solid Earth model with elastic compressible properties is employed. Figure 1 shows the position of the grounding line at year 2350 for these 3 runs for both SLIM (Fig.1a) and PLUS setups (Fig.1d). Projections for 2 intermediary setups are also shown. The first is identical to SLIM except that CESM2 SMB is used instead of the constant SMB (Fig. 1b). The second additionally includes PICOP in lieu of the linearly depth dependent ocean melt parameterization (Fig. 1c). Accross all setups, the inclusion of viscoelastic GRD effects reduces grounding line retreat by 30–200 km in 2350 (Fig. 1). Elastic stabilization plays a significant role in setups with depth-dependent ocean melt models, reducing grounding line retreat by 10–100 km (Figs. 1a, b). However, its effect is much weaker, with reductions of only 1–10 km, in setups using



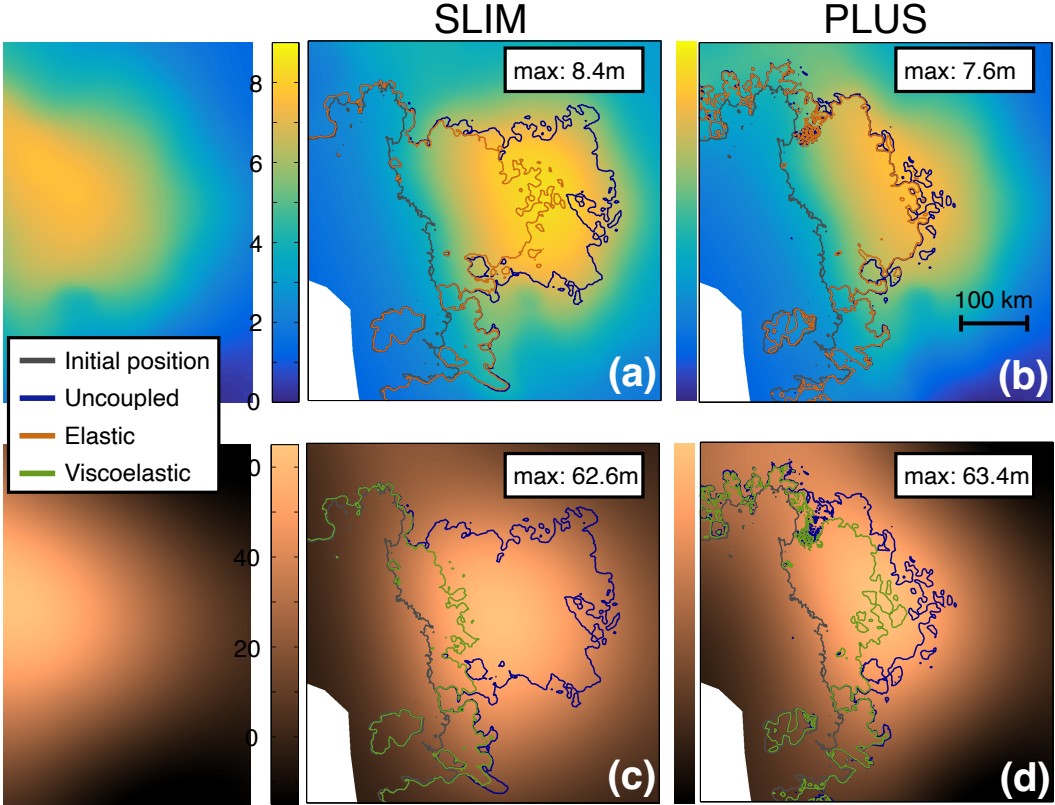

**Figure 2.** Bedrock uplifts at year 2350 for the SLIM (a and c) and PLUS (b and d) setups. The grounding line positions for the Viscoelastic (green), Elastic (brown) and Uncoupled (blue) runs are overlaid on top of the Elastic (a and b) and Viscoelastic (c and d) bedrock uplifts. The initial grounding line position is represented in gray.

PICOP (Figs. 1c, d). Overall, we observe that GRD effects induce delay in the grounding line retreat but barely any change in
grounding line retreat pattern (additional visualizations are available in appendix A2).

In contrast, ocean melt, SMB and mesh parameterizations not only affect the timing but also the pattern of the grounding line retreat. As expected, we observe a more extensive collapse in 2350 when including the aggressive CESM2 SP5-8.5 SMB instead of the constant SMB (Figs. 1a and b). On the contrary, the use of PICOP leads to a less extensive collapse compared to linearly depth dependent melt rate (Figs. 1b, c). Using an anisotropic mesh refined to 1 km in key regions instead of the 2 km
uniform mesh induces a more extensive collapse (Figs. 1c, d).

Figure 2 presents the amplitude of elastic and viscoelastic bedrock uplift in 2350 for both SLIM and PLUS setups. We observe 7.5 to 8.3 times more uplift in the Viscoelastic runs compared to the Elastic runs in the final year of our simulation (Figs. 3.a-d). For a given run type, both setups exhibit similar uplift amplitudes and slight differences in uplift patterns.

Figure 3 depicts the influence of GRD effects through time on Thwaites Glacier's GMSL contribution (a, b) and ungrounded
ice area (c, d) for both SLIM and PLUS setups. The ungrounded ice area, which represents the ice surface transitioning from





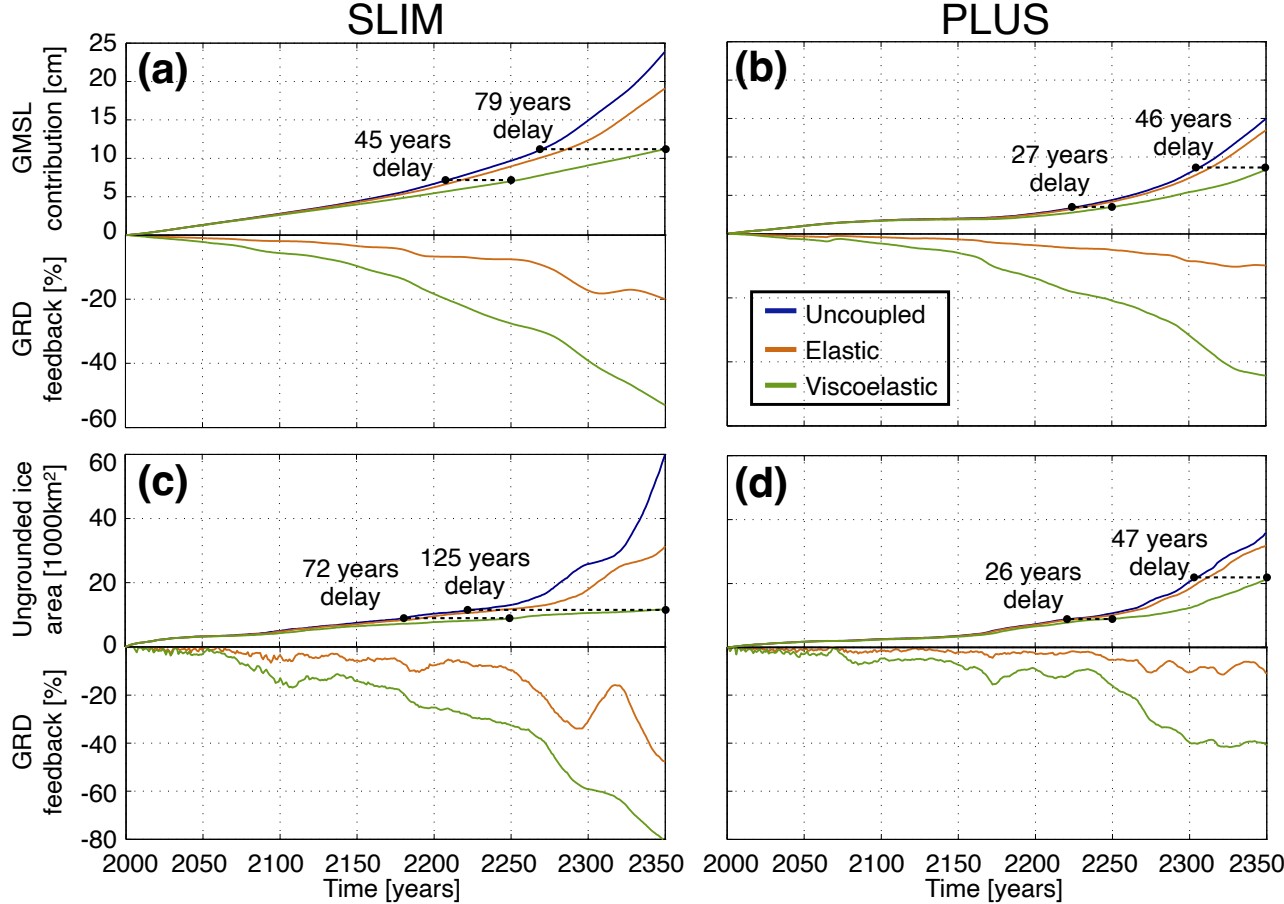

**Figure 3.** Thwaites Glacier's GMSL contribution (a and b) and ungrounded ice area (c and d) in the SLIM (a and c) and PLUS (b and d) setups. Cumulative values are plotted in the top halves of each graph while GRD feedbacks are plotted in the bottom halves. The GRD feedbacks correspond to the percentage (%) reduction in GMSL contribution or ungrounded ice area due to the incorporation of GRD effects. They are computed by comparing the results of the Viscoelastic (green) and Elastic (brown) runs with respect to the Uncoupled (blue) run.

grounded to floating during the simulation, serves as a proxy for grounding line migration. This metric is particularly relevant as GRD effects primarily delay grounding line migration with only slight alterations to its retreat pattern. We note that Haynes Glacier is included in GMSL contribution and ungrounded ice area calculations, with additional details provided in Appendix A3. The method used to compute the GMSL contribution, consistent with Adhikari et al. (2020), is also described in Appendix

A3. For the SLIM setup, viscoelastic stabilization produces substantial negative feedbacks on Thwaites' ungrounded ice area (12.9%) and GMSL contribution (5.6%) by 2100 (Figs. 3a, c). These feedbacks reach 80.7% and 53.1% respectively by 2350. In the PLUS setup these feedbacks are slightly smaller but nonetheless considerable. Notably, Thwaites' contribution to GMSL is reduced by 13.9% in 2200 and by 44.3% by 2350 (Fig. 3b). By 2350, GRD effects induce a 47 year delay for the PLUS setup and a 125 year delay for the SLIM setup on the grounding line retreat (Figs. 3c, d).



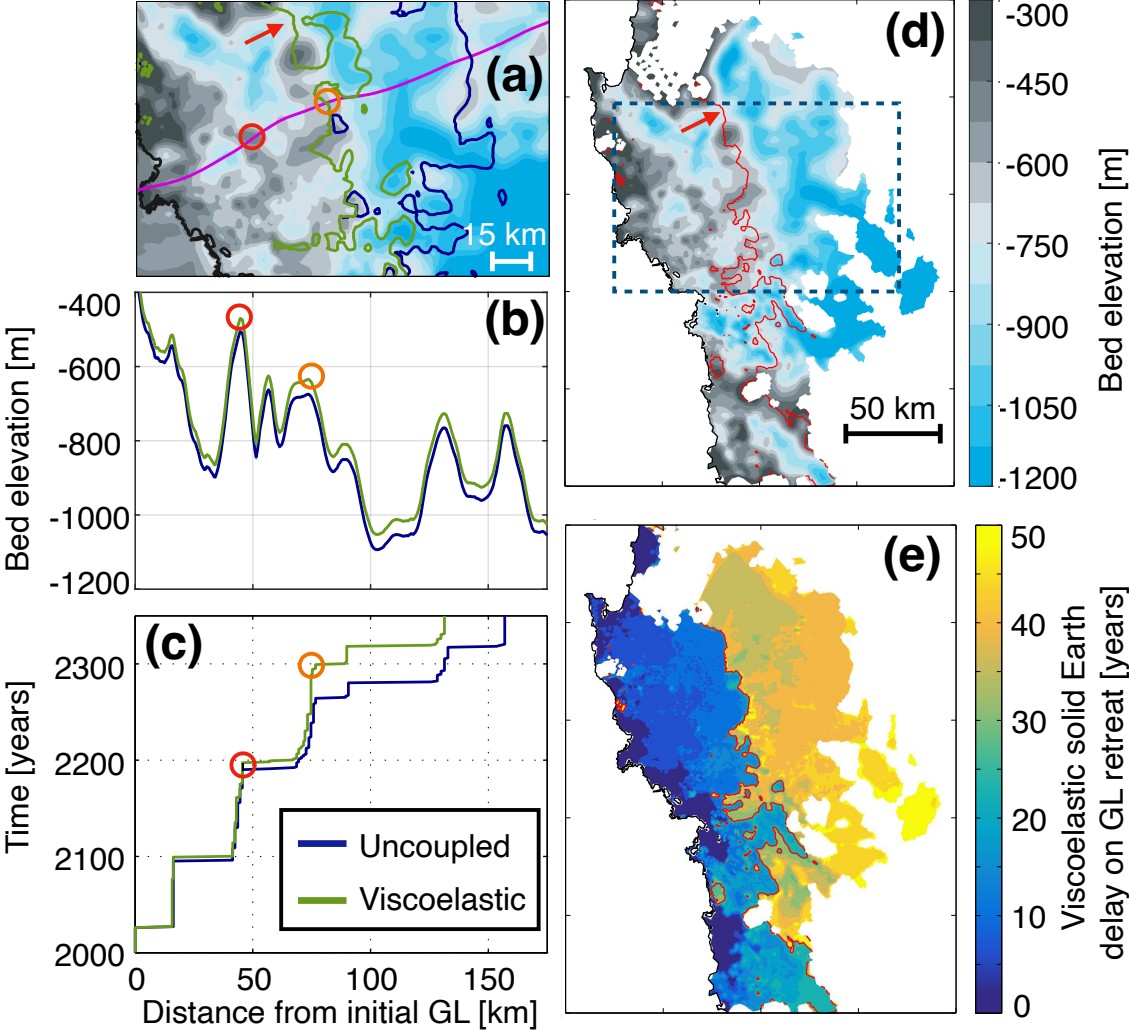

**Figure 4.** (a) Grounding Line (GL) positions at year 2292 for the Viscoelastic (green) and Uncoupled (blue) runs of the PLUS setup overlaid on the initial bedrock. Initial GL is represented in black. (b) Topography along the magenta flowline. The green line includes viscoelastic uplift at year 2292. (c) GL migration (x-axis) through time (y-axis). The x-axis is shared with (b). Circles indicate the locations of 2 key bedrock ridges which pin the GL for extended time periods. (d) Initial bedrock in the Viscoelastic run's GL retreat area. The blue dotted box corresponds to the area depicted in (a). (e) GL delay induced by GRD effects. The red line identifies the 25 year delay isocontour which coincides with the 2nd ridge of the basin and marks a sharp increase in the delay.

## 3.2 GRD effects prolong grounding line anchoring on bedrock ridges

For the Viscoelastic and Uncoupled runs of the PLUS setup, Figure 4a shows the position of the grounding line in Thwaites' basin at the year 2292 overlaid on the bedrock elevation. Figures 4b and 4c respectively depict the topography and the grounding





line position along the magenta flowline of Fig. 4a. We observe that the grounding line retreat evolves sporadically, alternating between short periods (∼ 1 year) of rapid retreat due to the retrograde bedrock slope, and long periods (decades) of quasi-static

anchoring on key bedrock features. A recent study highlighted the presence of two prominent ridges in the basin (Morlighem et al., 2020, Fig. 2a). The red circles (Figs. 4a-c) identify the first ridge that the grounding line reaches around 2100 in the PLUS setup. Here, we observe that the grounding line stays anchored 7 years longer in the Viscoelastic run compared to the Uncoupled run (Fig. 4c). The orange circles identify the second ridge (Figs. 4a-c) which anchors the grounding line for 70 years in the Uncoupled run and 100 years in the Viscoelastic run (Fig. 4c). The cause of this difference in grounding line behavior is

due to a larger bedrock uplift at the second ridge. This results from a larger ice mass change and a longer time period since the onset of ice mass loss, allowing more viscous deformation to accumulate.

Figure 4d shows the topography over which the grounding line retreats in the Viscoelastic run. Figure 4e represents the delay (number of years) separating the passage of the grounding line in the Uncoupled and Viscoelastic runs. The red line (Figs. 4d, e) represents the 25 year delay isocontour which coincides with the location of the second ridge (Fig. 4d). In Figure 4e it is

clear that an additional 20 to 30 year delay of the grounding line can be seen along the whole length of the second ridge, in agreement with results obtained through the study of individual flowlines (Fig. 4c) (Larour et al., 2019; Gomez et al., 2024). Another notable aspect of the interaction between the ice sheet and the solid Earth is highlighted by the constriction valley identified by the red arrow (Figs. 4a, d). Here, the grounding line stays pinned as well (Fig. 4a) despite low bed elevation due to the presence of the high lateral pinning points. This stabilization by 'proximity pinning points' is also extended by ∼30 years

(Fig. 4e) in the Viscoelastic run.

## 3.3 Impact of model resolutions on grounding line position and GMSL contribution

This study focuses on 3 spatiotemporal resolutions: (i) the spatial resolution of the ice sheet model, (ii) the spatial resolution of the GRD response (controlled by the Love number degree in ISSM), and (iii) the coupling interval between the GRD model and the ice sheet model. These resolutions determine the computational cost and the accuracy at which the bedrock is resolved

and updated throughout the simulations. Our goal is to determine the threshold values for these three resolutions that ensure the differences in GMSL contribution and ungrounded ice area remain under 5% throughout the Viscoelastic run, relative to the baseline Viscoelastic run presented in Figure 3 (green lines). We vary one parameter at a time, progressively degrading it while keeping all other parameters at their baseline values. Our results are summarized in Fig. 5.

The ice sheet spatial resolution corresponds to the resolution at which both the ice sheet and the underlying bedrock are

captured. We report the effects of coarsening the mesh by 1, 3 and 8 km. We keep the mesh structure unchanged for each setup. Hence, coarsening by 1 km means using a 3 km uniform mesh for SLIM. For PLUS, this means using an anisotropic mesh which progresses sequentially from 2–2.5–3 km in the grounding line retreat zone (Appendix A1). We observe a strong influence of the ice sheet spatial resolution through 2350. By 2100, coarsening by 1 km results in a 5.5% difference in ungrounded ice area for SLIM (Fig. 5c) and a 7.1% difference in GMSL contribution for PLUS (Fig. 5b), exceeding the 5% threshold in both

setups. In the SLIM setup, the GMSL difference to the baseline model reaches 63% for a 10 km mesh in 2200 and 12.9% for a 5 km mesh in 2300 (Fig. 5a). In the PLUS setup, coarsening the mesh by 3 km induces a 42.9% difference to the baseline in



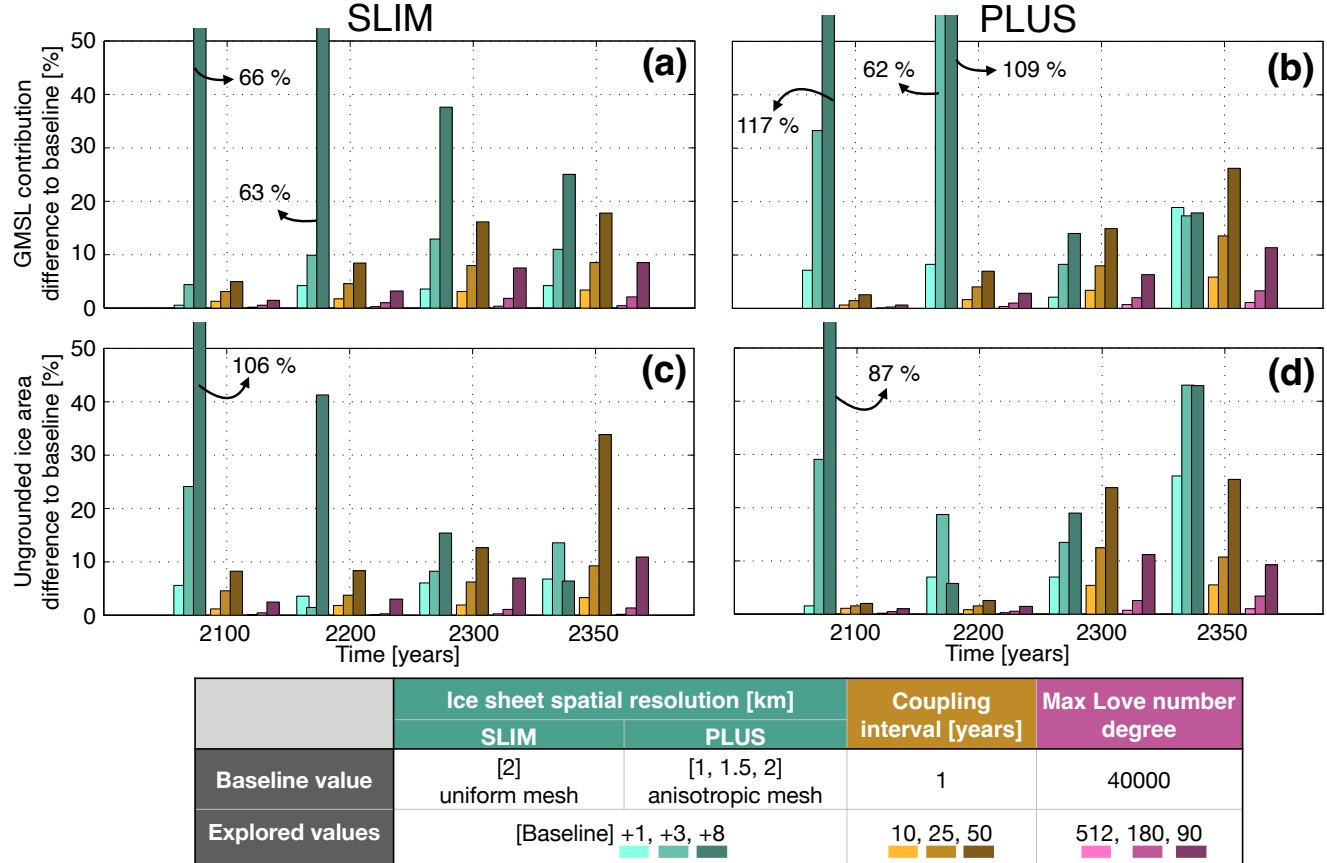

**Figure 5.** Sensitivity to (i) ice sheet spatial resolution, (ii) maximum Love number degree and (iii) coupling interval, for the SLIM (a and c) and PLUS (b and d) setups. Differences to the baseline in GMSL contribution are represented in the top row, and differences in ungrounded ice area are shown in the bottom row. Results are provided for four years (2100, 2200, 2300, 2350). For each parameter, three values are explored, with lighter colors indicating finer resolutions. For the ice sheet spatial resolution, the SLIM baseline uses a uniform 2 km mesh, while the PLUS baseline uses an anisotropic mesh refined to 1, 1.5 and 2 km in the region of grounding line retreat.

ungrounded ice area in 2350 (Fig. 5d).

The coupling interval is the number of years separating two GRD model computations which update the bedrock and sea-level positions. Using a 25 year coupling interval instead of the 1 year baseline leads to overestimating Thwaites Glacier's GMSL

contribution in 2350 by 9.2% in the SLIM setup and 13.6% in the PLUS setup (Figs. 5a, b). In the PLUS setup, a 10 year coupling interval induces over 5% differences in both ungrounded ice area and GMSL contribution by 2350 (Fig. 5d). In line with previous studies (Book et al., 2022; Larour et al., 2019), we find that the coupling interval must be kept below 10 years (Fig. 5b, d) to satisfy our 5% condition.

We define the spatial resolution of the GRD response as the minimum feature size that can be resolved by the GRD model.





In the ISSM, this is controlled by the maximum Love number degree 'n' which is the spherical harmonic degree at which we truncate our Green's function series. The corresponding minimum wavelength that can be resolved, accounting for the limit to prevent aliasing, is given by $\pi \frac{R}{n}$ where $R$ is the Earth's radius. We observe less than 1% differences in GMSL contribution and ungrounded ice area for both PLUS and SLIM setups when using $n = 512$ (39 km) (Figs. 5a-d). However, for $n = 90$ (222 km) the GMSL contribution differs by 8.5% from the baseline. Overall, we find that the GRD model's response can be resolved at

111 km ($n = 180$) and still verify our 5% condition through 2350.

## 4   Discussion

The results of this study indicate that the spatiotemporal resolutions of the ice sheet and GRD models can substantially impact projections of GMSL contributions and grounding line retreat. Here, we build on insights from analyzing coupled ice sheet and solid Earth interactions to explain this impact. We then discuss the implications of the observed GRD stabilization across

model setups.

### 4.1   Resolution requirements can be explained by coupled model mechanisms

Keeping all other components of the coupled model identical, substituting the 2 km uniform mesh for the anisotropic mesh (1 to 2 km in the area of grounding line retreat) leads to substantially more collapse in 2350 (Figs. 1c, d). In line with a growing body of literature, this suggests that specific grounding line dynamics can only be captured at kilometer scale resolutions

(Pattyn et al., 2013; Seroussi et al., 2014a; Seroussi and Morlighem, 2018; Robel et al., 2022). This is also consistent with the kilometer scale variations of the ridges and pinning points present in this area (Fig. 5a). A coarser mesh smooths out bedrock topography, potentially underestimating weaknesses in ridges like bedrock lows between pinning points (red arrow in Fig. 4a). We highlight here a possible explanation for the large differences observed in ungrounded ice area (25.9%) and GMSL contribution (18.8%) when coarsening the mesh of the PLUS setup by 1 km (Figs. 5b, d). We anticipate that in coupled

simulations, spatial resolution of the ice sheet model is twice as important. First, it conditions the capture of fine features which are reinforced by viscoelastic uplift. Second, its high sensitivity means it has a large influence on the amount of unloaded ice which in turn conditions bedrock uplift.

In the PLUS setup, differences in the ungrounded ice area remain small (< 2.5%) in 2100 and 2200 (Fig. 5d) for all coupling intervals considered. In 2300 a strong increase in these differences (5.3% for the 10 year interval and 12.5% for the 25 year

interval) can be seen. This increase can be explained by the fact that the grounding line breaks through the second ridge around 2290 (Fig. 4a). A similar effect can be observed in the 50 year interval simulation of the SLIM setup. In this case, differences in ungrounded ice area increase sharply from 12.6% in 2300 to 33.8% in 2350 (Fig. 4c). The longer interval allows the grounding line to un-anchor from the second ridge whereas the baseline model remains anchored in 2350 (see appendix A4 for visualizations). We conclude that longer coupling intervals lead to faster collapse as more time is provided for ungrounding to

happen prior to reinforcing key bedrock anchoring features. Results suggest that deformation on decadal timescale is significant enough to affect the floatability threshold. This is consistent with the relaxation timescales under 7 yr included in our transient




mantle rheology model.

A lower maximum Love number degree means that GRD deformation is applied more diffusely. For the PLUS setup, setting $n = 90$ results in over 12% less uplift (see Appendix A5 for details) in the vicinity of key bedrock features compared to the baseline, causing a 11.3% increase in GMSL contribution by the final year of the simulation (Fig. 5b). For both setups, we find that the maximum Love number degree can be set to $n = 180$ (111 km) and entail under 3.3% differences to the baseline (Figs. 5a-d). This suggests that the solid Earth response to Thwaites' mass loss is dominated by relatively large wavelengths on the order of the size of Thwaites' basin (Fig. 2) even when the Love numbers allow for responses on much smaller scales ($\sim 1$ km). In turn, this implies significant ice thickness changes on scales of 100 to 300 km (appendix A6). The resolution requirements identified here differ from previous estimates by Wan et al. (2022), which found bedrock deformation and geoid motions to converge within 5% for spatial resolutions of 7.5 km or higher in their GRD model. However, they reported that this threshold was influenced by short-wavelength ice mass loss patterns during the first 25 years of their simulation, which later transitioned to being dominated by longer-wavelength patterns. Some discrepancy may also arise from differences in the GRD models. For instance, their model uses grid-based GRD computations and incorporates lateral variations in rheology, whereas our model employs a Love number solver and accounts for transient mantle rheology. Finally, the metric used to determine convergence differs between the two studies. We emphasize that in this study, the spatial resolution of the GRD response refers to the smallest variations in ice and ocean loading that can be resolved by the GRD model. It is distinct from the resolution used to capture variations in mantle viscosity in 3D GRD models.

This sensitivity study is focused on Thwaites Glacier due to the low viscosity observed in West Antarctica and the importance of this glacier for future sea-level projections. We expect studies focused on other basins to obtain qualitatively similar results with varying numbers from those presented here as they depend on basin topography and mass change patterns. Additionally, we expect these numbers to vary based on the metrics used to determine convergence. To illustrate, the integrated metrics used in this study, such as GMSL contribution and ungrounded ice area, are well suited for studies focused on sea-level projections. However, these metrics may be less adapted for investigations focused on more localized variables such as vertical displacement from GPS stations (Kachuck et al., 2020). In such cases, the study of bedrock deformation is more relevant (Lucas et al., 2024). Nevertheless, this study provides a reproducible framework for analyzing the individual influences of various modeling choices. It can easily be applied to other convergence metrics, basins and modeling choices. Notably it will be useful to extend this work to ice rheology, basal friction and mantle rheological properties whose influences haven't been explored here.

## 4.2 Implications of GRD stabilization across different coupled models

In this study we investigate coupled ice sheet – GRD models and observe important viscoelastic stabilization across widely different model setups (Figs. 1 and 3). We conclude that in high-resolution simulations of West Antarctica, independently of other model components (mesh structure, ocean melt, SMB), rapid viscoelastic stabilization cannot be ignored. Indeed, uncoupled models risk overestimating Thwaites' contribution to GMSL by 5.6% in 2100 (Fig. 3a) and over 30% in 2300 (Fig. 3b). Our findings revisit prior conclusions based on the ISSM (Larour et al., 2019) by incorporating viscous GRD effects using





transient EBM rheology. They also complement previous studies that focused primarily on mantle rheology (Book et al., 2022; Gomez et al., 2024), confirming the need to include coupled ice sheet – GRD models in future projection frameworks.

In choosing only one set of mantle rheological properties we acknowledge that we are not exploring the uncertainty affecting the solid Earth structure. Hence, the results presented do not constitute definitive predictions of the state of the ice sheet, or the strength of GRD feedback. Rather, our focus lies in determining the spatiotemporal resolution requirement of coupled models,

for a given solid Earth parametrization that exhibits the decadal to centennial response required to explain current vertical land motion data in the region (Barletta et al., 2018). An aggressive yet realistic solid Earth parametrization is deliberately used here to study resolution sensitivity under a 'high-deformation scenario.' Hence, the requirements identified for coupling interval and spatial resolution of the GRD response are likely conservative estimates and should apply to less aggressive parametrization as well. This approach is particularly relevant given the large uncertainty in West Antarctica's mantle rheology (Ivins et al.,

255 2023).

This uncertainty in mantle rheology underscores the importance of ongoing investigations of 3D GRD models (Gomez et al., 2024). While our GRD model does not include lateral variations in mantle rheology, we expect similar conclusions for the resolutions studied here in coupled simulations using 3D GRD models given similar GRD feedbacks. Should GRD effects be weaker, we would expect resolution requirements identified here on coupling interval and spatial resolution of the GRD

response to be relaxed. We highlight here an opportunity for future studies using coupled models with 3D mantle rheology to investigate the distinct influences of (i) ice sheet spatial resolution, (ii) spatial resolution of GRD response, (iii) coupling interval and (iv) the spatial resolution of lateral variations of the solid Earth building on work by (Wan et al., 2022). We note that novel methods (Swierczek-Jereczek et al., 2023) must be further investigated to address the high computational cost associated with coupled models that incorporate 3D mantle rheology. We anticipate that studies focusing on computationally

intensive complex ice sheet dynamics (Pattyn et al., 2008; Favier et al., 2012) which seek to incorporate GRD effects will need to find compromises between the computational cost and complexity of GRD models. This is especially true when considering uncertainty quantification efforts which require large ensembles of model runs.

Finally, this study suggests that stronger GRD stabilization is to be expected in setups using depth-dependent basal melt parameterizations (Figs. 1a, b) compared to setups using plume-type basal melt models (Figs. 1c, d). This could be explained

by the fact that grounding line retreat in the former case is additionally driven by ungrounding ice pockets behind stabilizing ridges (depicted in Appendix A7). We hypothesize that instantaneous Elastic and rapid viscous uplift could delay the formation of these ungrounded ice pockets, providing a supplementary GRD stabilization mechanism which isn't present in setups using plume-type ocean melt models. Further research is needed to confirm this hypothesis.

## 5 Conclusions

We simulate the collapse of Thwaites Glacier through 2350, taking into account coupled ice sheet, solid Earth and sea-level interactions. We investigate the sensitivity of the coupled model to (i) the mesh resolution of the ice sheet model, (ii) the spatial resolution of the GRD response and (iii) the coupling interval between the ice sheet model and the GRD model. Results indicate



that coarsening a kilometer-scale ice sheet mesh by 1 km leads to significant changes in the projected GMSL contribution, with differences reaching 7.1% in 2100 and 18.8% in 2350. We find that both GMSL contribution and ungrounded ice area converge within 3.3% through 2350 for maximum Love number degrees above 180. This study also shows that decadal to sub-decadal coupling intervals are necessary to capture the full stabilizing effect of bedrock uplift.

Our findings point to an increasingly recurrent dilemma facing modelers who must ensure computational feasibility and cost without forgoing the capture of mechanisms which can only be resolved at high spatiotemporal resolutions. Indeed, high spatial resolution of the ice sheet, including bedrock topography, and short coupling intervals come at heavy computational cost. Notably, continental scale simulations cannot allow for the high resolution of local basin-scale studies. Further investigation of alternative GRD computational methods (Swierczek-Jereczek et al., 2023), mesh refinement techniques (Larour et al., 2012; Hecht, 1998; dos Santos et al., 2019) and other model optimizations (Han et al., 2022; van Calcar et al., 2023) are needed to solve this issue.

This study also highlights the crucial role played by pinning points and ridges in coupled simulations of the Amundsen Sea sector. We observe that modeled grounding line retreat alternates between phases of rapid migration between sets of laterally extending ridges and long quasi-static phases on these ridges. We find that viscoelastic adjustment could anchor the grounding line on these key bedrock features an additional 30 to 125 years. We expose this mechanism as a major driver of the enhanced stability observed in coupled simulations of marine ice sheets. Given the importance of this mechanism and the current uncertainty in bed topography (Morlighem et al., 2020), we emphasize the crucial need for ongoing and future bedrock measurements to better constrain the topography of subglacial ridges.

*Code and data availability.* The ISSM model used in this work is open-source and publicly available at https://issm.jpl.nasa.gov. The ISSM release version used for this work can be found on Zenodo https://doi.org/10.5281/zenodo.14548604. Output datasets and models comprise terabytes (TBs) of data and are stored on the Lou mass storage system at the NASA Advanced Super-computing facility at Ames Research Center. All datasets and code that was used to produce the output datasets and models are freely available on Zenodo https://doi.org/10.5281/zenodo.14548604

# Appendix A

## A1 Anisotropic mesh parametrization of the PLUS setup

We note that most of the grounding line retreat through 2300 occurs in the 1 km mesh area (light blue) for the Viscoelastic baseline run in PLUS. This is evidenced by the grounding line position at year 2300 (bright pruple). For reference, the Amundsen Sea sector region illustrated in (b) includes Pine Island, Thwaites, Smith, Pope, Kohler, and Haynes glaciers.

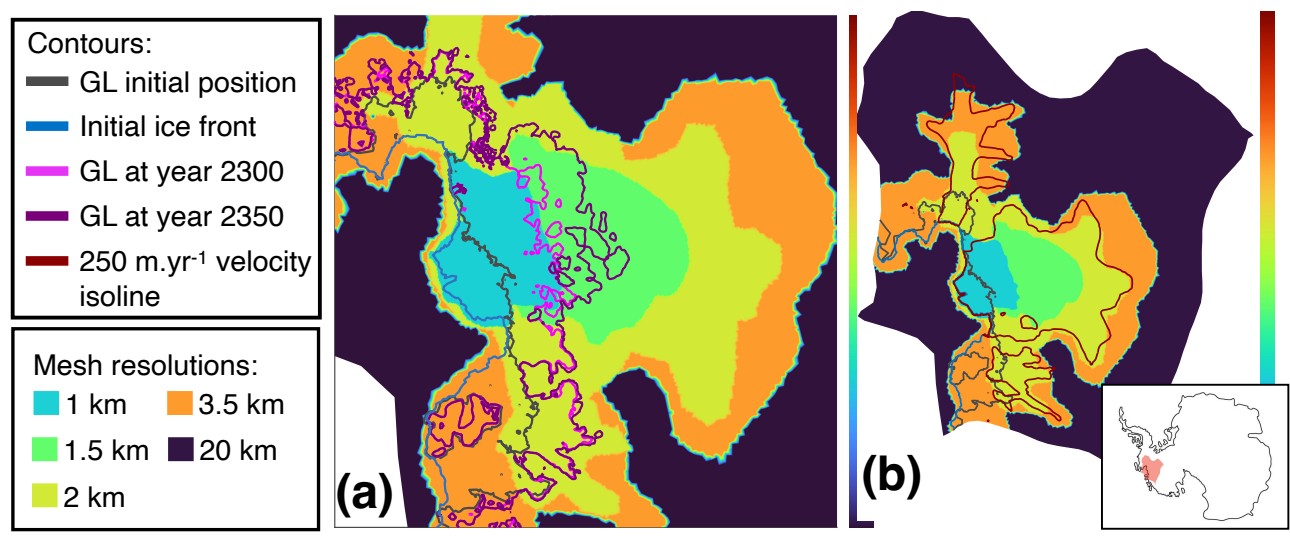

**Figure A1.** Illustration of the different areas of the PLUS anisotropic mesh with distinct mesh resolutions. Baseline resolution values are indicated in the legend. (a) Depicts the region shown in Figs. 1 and 2. The grounding line position at years 2300 and 2350 from the Viscoelastic run are overlaid for reference. (b) Shows the broader region of the Amundsen Sea sector on which this study focuses. The dark red contour represents the 250 m.yr$^{-1}$ ice velocity isoline. In both (a) and (b) initial grounding line and ice front positions are shown in gray and blue, respectively. The bottom right inset shows the location of (b) within Antarctica.





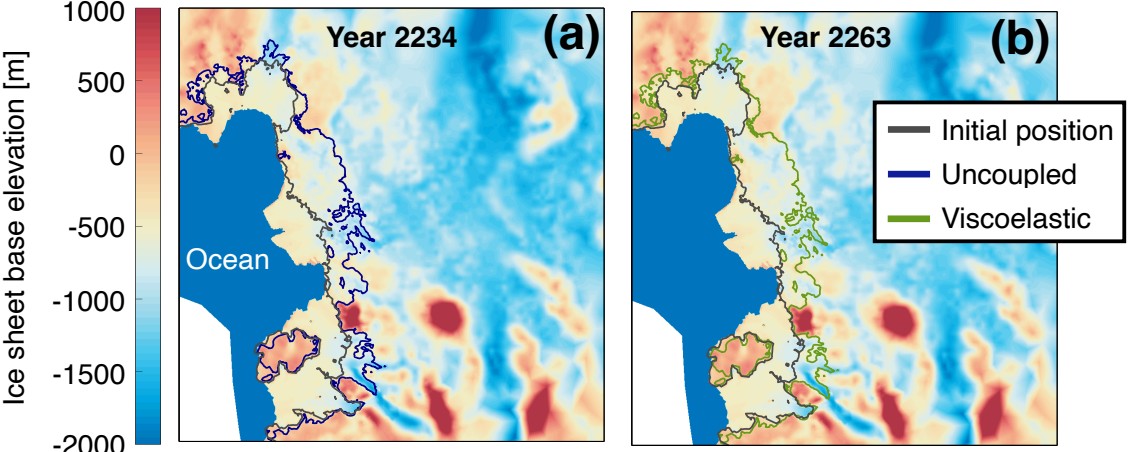

**Figure A2.** Grounding line projections in the Amundsen Sea sector in the SLIM setup. (a) Projection for the year 2234 in the Uncoupled run (blue). (b) Projection at year 2263 in the Viscoelastic run (green). Initial grounding line position is reported in gray.

## A2   GRD effects induce delay on grounding line retreat but barely any retreat pattern change



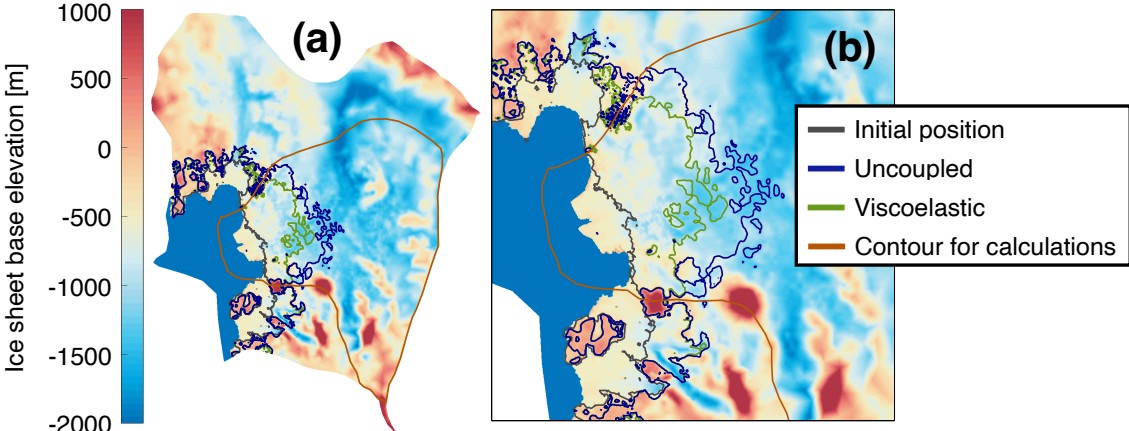

**Figure A3.** Contour of the region over which GMSL contribution and ungrounded ice area are calculated throughout the paper (brown). Viscoelastic (green) and Uncoupled (blue) grounding line projections in 2350 for the PLUS setup are overlaid for reference. Initial grounding line position is reported in gray. (a) Represents the region of the Amundsen Sea sector under study and (b) corresponds to the region depicted in Figs. 1 and 2.

## A3  Calculation details for GMSL contribution and ungrounded ice area

The brown contour roughly corresponds to the contour of the combined Thwaites and Haynes glaciers basins. It is slightly extended to fully capture the area in which we observe different grounding line positions for the Viscoelastic and Uncoupled
runs at the final year of the simulation. This contour defines a mask that identifies the elements and vertices of the mesh used to compute the GMSL contribution and ungrounded ice area.

We note that in coupled models, the classic "Volume Above Floatation (HAF)" method to calculate GMSL contribution is flawed due to bedrock and geoid motion. Here we present a brief overview of the method used to calculate GMSL contribution,
further details can be found in (Adhikari et al., 2020). We differentiate 3 regimes at each time step: (A) ice that remains grounded, (B) ice that becomes floating and (C) ice that remains floating. For (A), we simply use the variation in ice thickness to compute GMSL contribution. For (B), the contribution to GMSL mainly depends on the variations of the HAF accounting for bedrock and geoid motion. For (C), thickness variations of the ice only matter because of a density correction factor is needed to account for the discrepancies between ocean water density and freshwater density. We note that this density correction factor
is also taken into account in the cases (A) and (B).





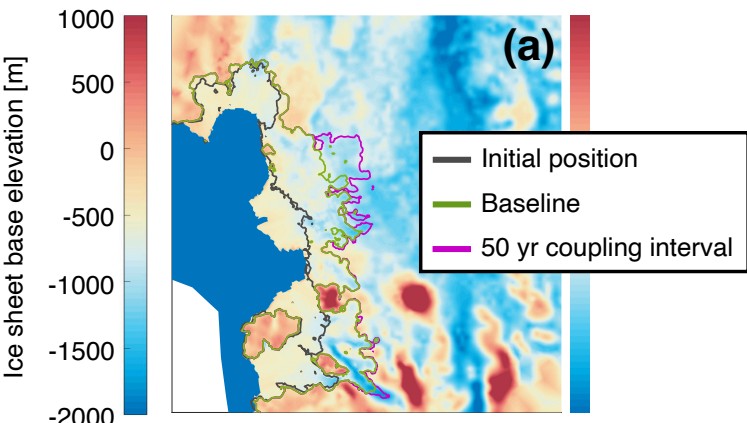

**Figure A4.** Grounding line projections in the SLIM setup at year 2350. The projection for the baseline Viscoelastic run (green) is shown alongside the projection for the 50 year coupling interval run (magenta). The initial grounding line position is reported in gray.

## A4 Increasing the coupling interval to 50 years causes the grounding line to un-anchor from the 2nd ridge in the SLIM setup





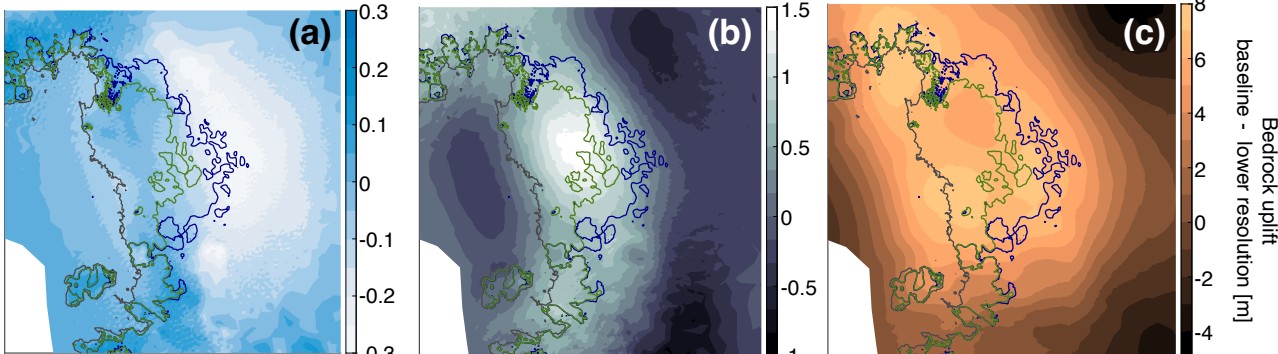

**Figure A5.** Loss in bedrock uplift due to setting the maximum Love number degree in the PLUS setup to (a) 512 (39 km), (b) 180 (111 km), and (c) 90 (222 km), instead of the reference value of 40,000. Grounding line positions at year 2350 for the Viscoelastic (green) and Uncoupled (blue) runs are overlaid for reference. The initial grounding line position is reported in gray. Note that the colorbar scales vary between panels.

**A5    Decreasing the maximum Love number degree reduces bedrock uplift in the grounding line retreat zone**



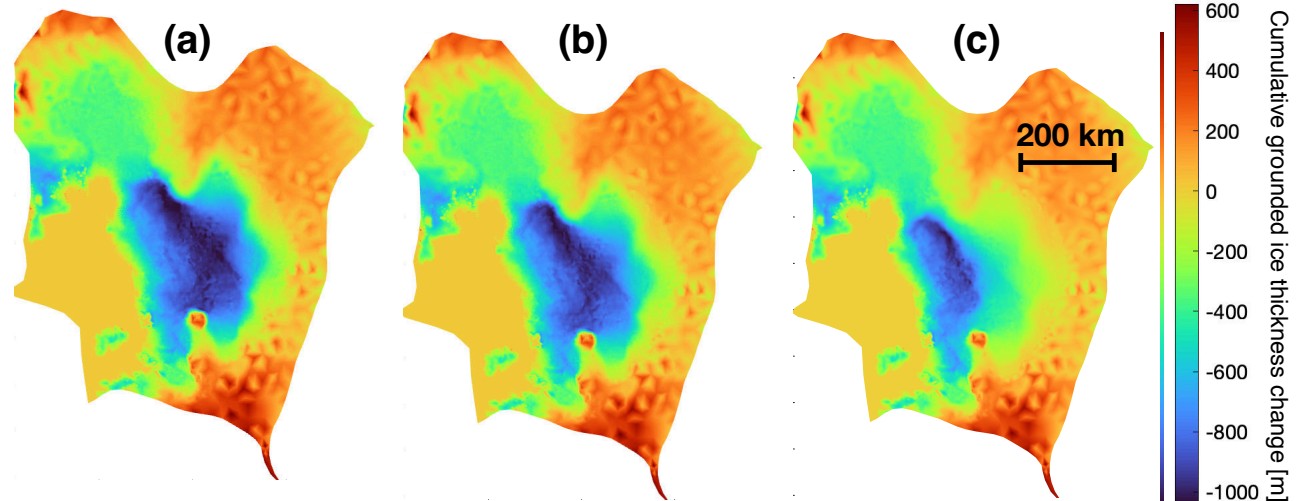

**Figure A6.** Cumulative grounded ice thickness change through 2350 in the (a) Uncoupled, (b) Elastic, and (c) Viscoelastic runs of the PLUS setup.

## A6 Grounded ice thickness change patterns are on the order of 100s of kms.




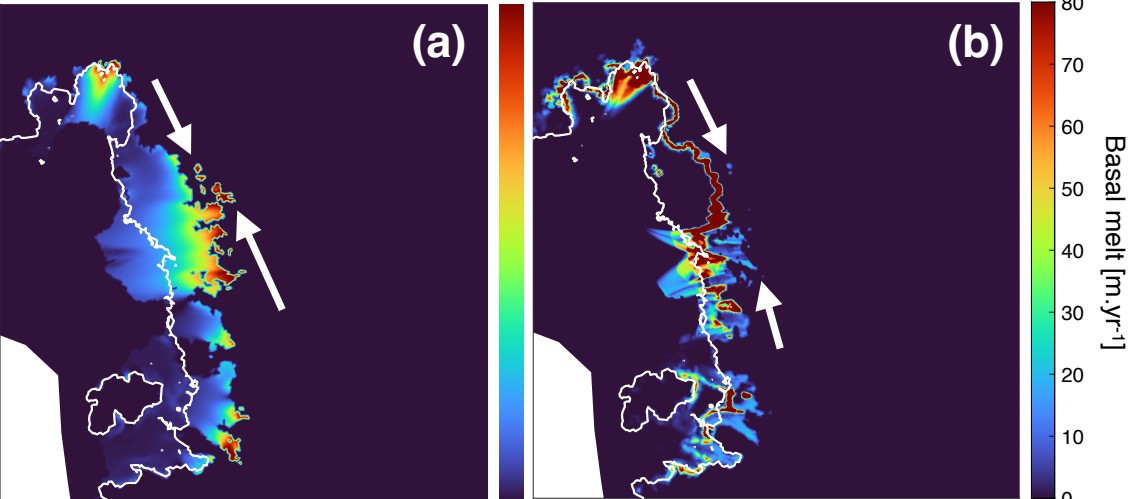

**Figure A7.** Ungrounding ice pockets (white arrows) behind the main grounding line. (a) For the SLIM setup (with depth-dependent basal melt), high melt rates occur due to the deep ice sheet base in these pockets. (b) For the PLUS setup (using PICOP), low melt rates are observed due to the shallow local base slope. The white contour shows the initial grounding line for reference. Note that the colorbar is capped at 80 m.yr$^{-1}$ for easier comparison, but in (b), basal melt rates reach a maximum of 280 m.yr$^{-1}$ at the grounding line.

## A7  Ungrounding ice pockets behind the main grounding line drive grounding line retreat in SLIM but not in PLUS

*Author contributions.* EL, LC and LH conceptualized the study. LH implemented the coupled model, ran the simulations and performed the analysis. LC provided solid Earth data and helped implement the coupled model. SA and EI provided feedback on the GRD model and effects. HS provided the initial ice model setup. TP provided ocean temperature and salinity data and helped implement PICOP. NS provided surface mass balance data and helped implement CESM2 climatology. LH wrote the majority of the main text with frequent and meaningful support from LC. All authors contributed to the writing and editing of the paper.

*Competing interests.* No competing interests to declare.

*Acknowledgements.* Part of the research was carried out at the Jet Propulsion Laboratory, California Institute of Technology, under a contract with the National Aeronautics and Space Administration (80NM0018D0004). It is supported by the following NASA Programs: Cryosphere: Ice Mass Balance Intercomparison Exercise - Phase 3 (Grant #509496.02.08.13.79), Earth Surface and interior: Ice Sheet Collapse and Soft Mantle Rheological Response (Grant #281945.02.03.13.11), Modeling Analysis and Prediction: Mass transport driven coastal sea level (Grant #509496.02.08.12.39), NASA Sea-Level Change Team: From grounding lines to coastlines: an integrated approach to barystatic



sea-level projections (Grant #281945.02.47.05.18), New Investigator and Early Career Program (20-NIP20-0030) and GRACE-FO Science
Team (19-GRACEFO19-0001). LH acknowledges support from a Stanford Mechanical Engineering Department Fellowship, a Stanford Civil
and Environmental Engineering Department Fellowship and the Stanford Data Science Scholars. HS acknowledge support from the Novo
Nordisk Foundation under the Challenge Programme 2023 - Grant number NNF23OC00807040. Computing resources supporting this work
were provided by the NASA High-End Computing (HEC) Program through the NASA Advanced Supercomputing (NAS) Division at Ames
Research Center. The authors also acknowledge the support of the ISSM development team and the feedback from NASA JPL's Sea Level
and Ice group.



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
