# Peer review of "Reinforced ridges in Thwaites Glacier yield insights into resolution requirements for coupled ice sheet & solid Earth models"

_EGUsphere, 2024_

## Referee Comment (RC2)

**General comments**

In the present work, the authors study how the grounding line retreat of Thwaites Glacier, as predicted until 2350, is influenced by the spatial resolution of the ice sheet model, the GRD model, and by the temporal resolution used to couple them. In particular, they vary these resolutions independently from each other and observe significant differences of the forecasted GMSL contribution, which mostly depends on the ice sheet resolution for short prediction horizons (2100), and on a combination of the resolution parameters for longer horizons (2350). The observed differences are explained by the fact that high resolution runs capture bathymetric features more accurately, which can be enhanced by GRD effects. The latter is well exemplified by showing how the grounding line retreat varies if the isostatic adjustment is neglected, purely elastic or viscoelastic.

I believe that the manuscripts submitted by the authors is a valuable contribution to the literature, since it provides a more "orthogonal" analysis of the three main resolution parameters than what was made to date. I appreciate that many permutations of the model setup were made to highlight the main sources of difference in future GMSL contributions and I largely agree with the mechanistic explanations given in the paper, which are well supported by the figures. Overall, I enjoyed reading the article, which is well written, instructive and correctly embedded in the present research context.

Before a possible publication, I therefore only envision minor revisions, which are listed below. I wish all the best to the authors for the rest of the publication process and look forward to reading the final version of the manuscript.

Best regards,

Jan Swierczek-Jereczek

**Specific comments**

1. The title is somewhat vague: you don't mention the exploration of various resolutions, which is the central contribution of the work. I believe something like "Spatiotemporal model resolution significantly affects the interaction between the solid Earth and Thwaites Glacier" is more coherent with your storyline. I understand that you want to mention ridges because they are the explanation you find for this sensitivity… but "Insights from Reinforced Ridges" does not convey this adequately in my opinion.

2. Your motivation could benefit from a clearer thread, which I try to convey in the forthcoming sentences: ISMIP6 models forecast that grounding line retreat in the Amundsen Sea Embayment (ASE) will lead to the greatest GMSL contribution over the coming centuries. In this region, the upper-mantle viscosity is particularly low and implies that the GRD response, which provides stabilising effects on grounding line retreat, is particularly fast. The stabilising effects of GRD can be enhanced by the presence of ridges and confinements, which have been identified in ASE but can only be represented by using high model resolutions. This is your central motivation to

study the resolution dependence, and deserves to be highlighted in a clearer way. This is particularly the case for the abstract, which I invite you to change accordingly.

3. L. 11: "extends buttressing" → "delays grounding line retreat".

4. L. 18: "In the fastest melting region of the ice sheet, Thwaites Glacier has displayed signs of early collapse (Joughin et al., 2014)." → "In Thwaites Glacier, the fastest flowing region of the ice sheet (Rignot et al., 2014), early signs of a collapse have been identified (Joughin et al. 2014, van den Akker et al. 2025), although this is contrasted by other studies (e.g. Hill et al. 2023)". I include all additional references at the end of this document.

5. L. 24: I believe this is the good moment in the paper to mention that, in the context of Antarctica, the order of importance of GRD feedbacks is: first deformation, second gravity and last rotation, the latter being almost negligible due to the high latitude. I think this would help the (non-expert) reader. Alternatively, you can mention this in the discussion.

6. L. 30: It feels much more accurate to say that the mantle rheology controls t*he time scale* of GRD effects (particularly the deformational aspect, which might be nice to emphasise), not the "strength" of it. I suggest correcting this in other places too (see below).

7. L. 49: "Ice sheet models have several major components, including mesh structure, basal melt and surface mass balance (SMB) parameterizations, which may affect the sensitivities of coupled simulations to (i), (ii), and (iii). To account for such effects, we report our results for 2 widely different coupled model setups labeled SLIM and PLUS. These have the same GRD model but different ice sheet models representative of the spectrum of complexity of modern ice sheet models." → "Ice sheet models are subject to a wide number of modelling choices, which can greatly affect the sensitivity of the coupled simulations. To check the robustness of our assessment of the sensitivity to resolution, we report our results for 2 widely different coupled model setups, labeled SLIM and PLUS, which differ in the imposed surface mass balance, the basal melt law and the mesh type."
I believe this captures a bit better your motivation: you introduce SLIM and PLUS because you want to show that the importance of the resolution does not depend on other key modelling choices (SMB, BMB and mesh type). It is in fact what you observe, since SLIM and PLUS look qualitatively the same in Figure 5. I feel like the latter is a very strong message of your paper and think it could be stated more emphatically.

8. L. 55: How is warming applied to the ocean and to the atmosphere in SLIM? Are they just uniform anomalies? Please specify how you derive that from SSP5-8.5!

9. L. 64-65: This feels like a particularly low value of the model time step… Did you observe significant changes when using higher values? This might be important to discuss since the article is all about spatiotemporal resolutions.

10. L. 66-67: Are you optimising for each of the model setups individually? This question applies to all the permutations of the paper (SLIM / PLUS, uncoupled / elastic / viscoelastic, varying resolutions). This is important to mention somewhere in the manuscript. Also, could you include a figure in the appendix showing the error with respect to present day after the optimisation? That would be very helpful.

11. L. 68-69: What do you gain from imposing this GRACE extrapolation? Does it change a lot from just imposing present-day thickness? From the reader's perspective, it is unclear how this can contribute to a more accurate experimental setup. In my opinion this is quite superficial since, if Thwaites is really the first to collapse, it is unlikely to be affected by what will happen in neighbouring Glaciers.

12. L. 78: "The Extended Burgers Material (EBM) (Ivins et al., 2022) rheology is…" → "As described in the methods, the Extended Burgers Material (EBM, Ivins et al., 2022) rheology is…".

13. L. 105: "by the transient EBM rheology described in the Methods section, we compare…" → "by the transient EBM rheology, we compare…"
You actually don't describe the EBM rheology in the methods and should briefly do so, since it is important to understand the parameters of Table 1.

14. L. 102: The subsection title is quite misleading since it suggests that there is a single baseline setup, although you have two of them. Please change this.

15. Figure 1: "+ PICOP" → "+ PICOP & CESM ocean". Using the CESM ocean fields here might actually be the modification that has the largest impact! For instance, does CESM represent the warm-water intrusion observed over the last decades in ASE? I suspect not since this is quite difficult, but that of course leads to a massive reduction of the grounding line retreat! This could drastically change the statement made at l. 268-273, which I think is largely unfounded: from what I understand, the linear melt law depends on the depth of the ice shelf base, which is largely unaffected by GRD. I believe the driver here is rather the CESM ocean forcing and, unless you prove the contrary, I would remove Appendix 7 and show a difference map of the ocean forcing instead.

16. Figure 2: I am quite surprised that the comparatively small elastic bedrock uplift (max 8 m) has such a strong influence on SLIM (and SLIM + CESM2 SMB), as shown in Figure 1. The change in grounding line position between uncoupled and elastic run is massive, whereas it is barely noticeable as soon as you use PICOP. This might be related to the previous point…. Alternatively, it might be due to how you apply melt at the grounding line, which you really should mention somewhere here.

17. L. 117: "As expected" → I wouldn't expect the SMB of CESM2 SSP5-8.5 to provoke a grounding line retreat, since it probably presents a substantial increase in precipitation, while maintaining surface temperatures that are largely below 0°C. I guess your answer to Point 8 might clarify this difference. If not, please provide an explanation.

18. Figure 4 and the corresponding text explanations are particularly nice!

19. L. 160 and following: I understand that defining an error threshold is particularly convenient for the analysis since it allows a simple statement like "resolution sufficient" or "resolution insufficient" (and it also allows a comparison to Wan et al., 2022). However, I believe that such a statement depends on the application case and I would therefore recommend phrasing the rest of the results in a less binary way (which does not prevent a comparison to Wan et al., 2022). For instance, an error of 6% is totally acceptable in almost all cases, especially given that there are many additional sources of error (the laterally variability of Earth properties for instance). In particular, this would prevent sentences like "Overall, we find that the GRD model's

response can be resolved at 111 km (n = 180) and still verify our 5% condition through 2350." (l. 184), which, as you know, is completely wrong for laterally varying viscosities. Please discuss this sentence if you want to keep it! Along this line, it should be highlighted that n=180 (at l. 280) is a lower bound on the Love number degree, since you have a 1D Earth.

20. L. 165 and following: you only mention differences in GMSL and ungrounded area, without specifying the sign of it (some exceptions as e.g. l. 174). This is of course conditioned by the fact that you don't include this information in Fig. 5 either. Including the sign in both the description and the figure would make everything much easier to read!

21. L. 167 and following: you begin describing the results with a lot of detail. I feel like an overall picture prior to that would be welcome. Something like: "For SLIM as well as for PLUS, coarsening the resolutions invariantly produces a qualitatively similar reduction of accuracy. This effect is dominated by the ice sheet model resolution for short prediction horizons (2100), whereas the GRD resolution and the coupling time step become similarly important for longer horizons (2350). In particular…"

22. L. 192-199: You might want to cite Williams et al. (2025), which is the latest work related to this.

23. Figure 5: It is clear why the error would increase over time but... can you explain why it decreases over time in the case of the ice sheet model? I guess the *absolute* error increases but, because the GMSL contribution increases faster, and this leads to a decrease of the *relative* error. Can you confirm? If yes, please mention this in the text.

24. Figure 5: even the highest resolution used here has not converged since it displays significant differences in GMSL contribution with e.g. +1km for the ice sheet model resolution. This is unsurprising and completely ok but needs to be stated!

25. In almost all maps of Thwaites that you show, you have a white cut-out in the bottom left corner. This could easily be fixed and would make everything a bit prettier.

26. The mask shown in Fig. A3 looks very arbitrary. I don't understand this choice, which does not look like it is motivated by a drainage basin. Please explain this.

**Technical corrections**

1. You use "Amundsen Sea sector" a lot. I would suggest using Amundsen Sea Embayment (ASE), which is becoming a standard acronym.

2. L. 2: "century timescale" → "centennial timescale". Also at line 15.

3. L. 22-23: "and more recently centennial (...) timescales" → "and more recently centennial timescales (...)." Also, you should add Gomez et al. (2024) here, which, to my knowledge, is the latest article that supports this point (especially since you already have it in your refs).

4. L. 23-24: "In the vicinity of the grounding line –the region in which grounded ice becomes afloat, these interactions mainly comprise stabilizing negative feedbacks from sea-level fall and bedrock uplift" → "In the vicinity of the grounding line, the

region where grounded ice becomes afloat, these interactions mainly comprise negative feedbacks on grounding line retreat. They consist in a gravitationally-driven reduction of the regional sea-surface height and a bedrock uplift due to the reduction of the surface load applied on the solid Earth."

5. L. 25: "The negative feedbacks may be regionally enhanced by low mantle viscosity" → "In ASE, the deformational feedback is faster than the global average, due to upper-mantle viscosities that are particularly low." Here, I believe you should cite Ivins et al. (2023) (which you already have in your refs) rather than Coulon et al. (2021) since the former infers upper-mantle viscosities whereas the latter is a simulation study based on relaxation times.

6. L. 28: "Glacial Isostatic Adjustment –GIA" → "Glacial Isostatic Adjustment – GIA"

7. L. 31: "the strength of GRD effect" → "the timescale of deformational effects". (Of course this has repercussions on gravity and rotation, which, however, are of second order). Also, this is a much better place to cite Coulon et al. (2021)!

8. L. 59: "sub ice shelf water" → "sub-shelf meltwater"

9. L. 68: "the model's mesh cover…" → "the model mesh covers…"

10. L. 78: "(Adhikari et al 2016)" → "(Adhikari et al., 2016)"

11. Table 1 and throughout the document: the multiplication between units is represented by "." (low dot) when it should be a "·" (middle dot).

12. Table 1: as you mention in the discussion, the viscosity of the asthenosphere and the upper mantle are rather in the lower range of literature values. I think this should be mentioned earlier because it conditions the interpretation of all the results you present: the differences between elastic and viscoelastic basically represent an upper bound!

13. Table 1: I would make two distinct columns for \tau_H and \tau_L.

14. L. 97: "appendix" → "Appendix". Also at lines 115, 219.

15. L. 102 and following: I understand you write "Viscoelastic", "Uncoupled" and "Elastic" with a capital letter because you consider them to be substantives. I would be in favor of just writing them as simple adjectives (viscoelastic, elastic and uncoupled runs). This feels lighter and equally well understandable.

16. Figure 1: I think a different colorbar would greatly improve the visualisation of the bedrock elevation (by the way, why do you prefer to use "ice sheet base elevation" in your colorbar label?). For instance, something like what is used in Garbe et al. (2020) could be nice (with adjusted range):

[Figure]

17. All figures: the effort of using a large font for legibility is well appreciated but you might have overdone it a bit. The difference to the font size of the preprint is quite large sometimes (assuming the figures will have a roughly similar size in the final version).

18. L. 110: "depth dependent" → "depth-dependent". I think the dash is helpful here since you are concatenating many substantives.

19. L. 113: "SP5-8.5" → "SSP5-8.5"

20. L. 121: there is a weird indentation here. Also at lines 124, 164.

21. Figure 3 is very convincing but would benefit from having separate panels for GMSL contribution, ungrounded area and GRD feedback (right now the y-axes are very confusing). I would suggest something like:

[Figure]

22. Figure 4 is also great albeit improvable. It would benefit from having a larger orange circle in panel (a), which corresponds better to the ridge. Since the ridges are spatial regions, red and orange vertical shadings in panels (b) and (c) are much more appropriate than the circles used in the current version. I would welcome a similar plotting style for panels (b) and (c) (right now they present different frame widths and line styles for the grid). "GL delay induced by GRD effects" → "Delay of GL retreat due to GRD effects". Please make the colorbar of panel (e) discrete instead of continuous in order to match the map. Finally, panel (d) could be deleted altogether since it does not provide useful information apart from the rectangle delimitation and the arrow, which could also be easily placed in panel (e). If you decide to keep panel (d), please remove the white mask beyond the GL retreat, which actually only applies to panel (e).

23. L. 148-149: remove "The red line (Figs. 4d, e) represents the 25 year delay isocontour which coincides with the location of the second ridge (Fig. 4d)." You already mentioned that in the figure caption.

24. L. 162: "degrading" → "varying"

25. L. 180: "Love number degree 'n' which is the…" → "Love number degree, $n$, which is the…".

26. L. 226: "that in this study, the" →"that, in this study, the"

27. L. 236: "Nevertheless, this study provides a reproducible framework for analyzing the individual influences of various modeling choices." I think you should rephrase this

since testing parameters independently from each other is nothing conceptually new. In my opinion, you are not providing a framework but just complying with good scientific practice to isolate the individual parameter sensitivities.

28. L. 245: "based on the ISSM" → "based on ISSM". More coherent with how you treat acronyms like SLIM, PLUS, PICOP....

29. L. 251: "An aggressive yet realistic solid Earth parametrization is deliberately" → "A fast yet realistic parametrization of the viscoelastic uplift is deliberately"

30. L. 252: quotation marks look weird, please correct.

31. L. 264: (Swierczek-Jereczek et al., 2023) → (Swierczek-Jereczek et al., 2024). You are citing the preprint although there is a published version.

32. L. 265: "on computationally intensive complex ice sheet dynamics" → "on computationally expensive ice sheet dynamics"

33. L. 313: "Volume Above Floatation (HAF)" → "Height Above Floatation (HAF)". I am not sure this section really brings useful information compared to simply referring to Adikhari et al. (2020).

34. In the references, some acronyms are written oddly: e.g. "sE", "gMD", "tC"...

**Additional references**

1. van den Akker, T., Lipscomb, W. H., Leguy, G. R., Bernales, J., Berends, C. J., van de Berg, W. J., and van de Wal, R. S. W.: Present-day mass loss rates are a precursor for West Antarctic Ice Sheet collapse, The Cryosphere, 19, 283–301, https://doi.org/10.5194/tc-19-283-2025, 2025.

2. Hill, E. A., Urruty, B., Reese, R., Garbe, J., Gagliardini, O., Durand, G., Gillet-Chaulet, F., Gudmundsson, G. H., Winkelmann, R., Chekki, M., Chandler, D., and Langebroek, P. M.: The stability of present-day Antarctic grounding lines – Part 1: No indication of marine ice sheet instability in the current geometry, The Cryosphere, 17, 3739–3759, https://doi.org/10.5194/tc-17-3739-2023, 2023.

3. Williams, C.R., Thodoroff, P., Arthern, R.J. *et al.* Calculations of extreme sea level rise scenarios are strongly dependent on ice sheet model resolution. *Commun Earth Environ* **6**, 60 (2025). https://doi.org/10.1038/s43247-025-02010-z

4. Swierczek-Jereczek, J., Montoya, M., Latychev, K., Robinson, A., Alvarez-Solas, J., and Mitrovica, J.: FastIsostasy v1.0 – a regional, accelerated 2D glacial isostatic adjustment (GIA) model accounting for the lateral variability of the solid Earth, Geosci. Model Dev., 17, 5263–5290, https://doi.org/10.5194/gmd-17-5263-2024, 2024.

---

## Author Comment (AC1)

In this paper, the authors use an ice sheet model coupled to a Gravitation, Rotation, and Deformation (GRD) model to investigate the impact of modeling choices on the retreat of the Thwaites glaciers under warming scenarios. They conclude that resolving the grounding line of the ice sheet model has the strongest impact on sea level change estimation compared to the GRD model and the coupling interval. However, they highlight the importance of including a viscoelastic response of a GRD model with a coupling interval of at least 10 years is needed to prevent an overestimation of Thwaites contribution to sea level change.

I would like to thank the authors for the particular care they took in submitting a manuscript that is well written with a sound layout and methodology, and very nice figures. I would encourage its publication after minor revisions.

I will begin with general comments followed by specific ones.

We thank the reviewer for his positive remarks about our manuscript, its findings and its figures. In the sections below we address each of the reviewer's comment and detail the way in which the manuscript has been modified to incorporate the reviewer's feedback. We sincerely thank the reviewer for his comments, both general and specific, which have noticeably improved the quality of the manuscript.

**General comments**

I think section 2.1 or the description of the ice sheet models could use additional information.

From what I could read, you do not mention the basal sliding law you are using, nor do you mention whether you are using a grounding line parameterization to diagnose the grounding line. You mention a floatation threshold which is vague. In addition, please add how you are applying ocean melt at the grounding line.

Thank you for this comment, section 2.1 has been modified accordingly, and an Appendix 2 has also been added to provide further details.

I also think that it would be useful to say a few words on how you are performing your model initialization and the dataset you are using for this (e.g., geothermal heat flux, bed topography, …) and at the very least add a figure on the ice sheet initial state (error w.r.t velocities since this is the metric you are choosing) and the drift of your initial state prior to applying the SSP5-8.5 scenario forcing.

Thanks for this suggestion, we have added this information and these figures to the manuscript and Appendix 2.

While the differences in SLIM and PLUS are described, in section 2.1, their differences are fundamental in the rest of the study. For this reason, I would find it useful to have a table in that section summarizing the differences between both allowing the reader not to have to find the detail in the text.

Thanks for this suggestion, we have added this table (table 1) to the manuscript.

In section 2.2, please add a figure of the GRD initial state and the bed uplift drift from the GRD model prior to adjusting the ice sheet model.

Initially, as no ice variation is fed into the GRD model, there is no bed uplift drift of the GRD model. The initial state of the solid Earth is simply the initial bed topography which can be visualized in Fig 1.

I am not sure there is a need for section 2.3. Lines 94-97 belong to section 2.1 and the resolution of the GRD model would fit in section 2.2. In addition, please add whether the ISSM grid is evolving over time or remains fixed throughout the experiment.

We have incorporated section 2.3 into sections 2.1 and 2.2. The ISSM grid does not evolve over time and remains fixed throughout the experiment. We have added this information in Appendix 1 where we felt it was most appropriate.

**Line by line comments**

Line 9: replace "difference" by increase or decrease.

The abstract has been modified accordingly.

Line 48: does this mean that the ISSM and GRD models have grid nodes in common? If not, please add how you perform the regridding from one model to the other.

Thanks for this comment. This does indeed mean that the ice sheet and GRD models share the same grid nodes. We've updated the manuscript to include this information.

Line 50: what do you mean by "SMB parameterization"? In the text you clearly state you are using a climatology form RACMO and forcings from CESM2. These are not parameterization but rather forcing choices.

Manuscript modified accordingly.

Line 65: "2 weeks to capture …" is it really to capture the rapid changes or is it also necessary to avoid a CFL violation? Maybe both?

Thank you for the comment. The 2-week time step was chosen both to capture rapid changes and to satisfy the CFL condition. This is a conservative choice to ensure that time stepping does not interfere with our analysis of other model parameters, such as spatial resolution. Indeed, preliminary tests at project inception showed that longer time steps (1–2 months) had little to no impact on early simulations. We have updated the manuscript to reflect this information.

Line 68: Please explain this sentence further. What do you mean by icecaps outside of the ASE, do you mean around the world or just in Antarctica? (The previous sentence mentions that your grid covers the entire globe.) Also, if your goal is to focus on the ASE, what is the added value of using GRACE's trends for SMB outside of the ASE as opposed to keeping the SMB to the value used at initialization?

Thanks for this comment. This sentence refers to icecaps around the world (including but not limited to Patagonia, Greenland, High Mountain Asia etc.) The added value of using Grace data is to account for ocean loading due to far-field ice mass change (Gomez 2020), notably in the first few decades of the simulation. The idea was to include any possible effects of present-day rising sea level on WAIS initial stability. In the long-term, contributions from other glaciers are dwarfed by that of WAIS of course. We've modified the manuscript to include this information

Line 89: replace "(Ivins et al. 2020 …)" by "Ivins et al. (2020, 2023) and Lau and Faul (2019)".

Manuscript modified accordingly.

Line 99: what do you mean by "major coupled model components"?

Manuscript modified accordingly to avoid confusion. The major coupled model components referred to those listed in the following paragraph: "GRD effects, mesh structure, basal melt parametrization and SMB forcings."

Line 104: same comment as in line 50.

Manuscript modified accordingly.

Line 194: If by "kilometer scale" you mean ~1km, please rephrase as Leguy et al. (2021) showed that the needed resolution to capture grounding line behavior is model dependent.

The manuscript has been modified accordingly.

Line 200: please replace "twice" by another comparative word (maybe "more"). Until prove such a number, you cannot really claim it.

Manuscript modified accordingly.

Line 237-238: Berdahl et al. (2023) showed that variation in basal sliding laws influences the rate of retreat of the grounding line and, in the presence of GIA, the collapse of Thwaites could be reversed.

Thanks for pointing this paper out, the manuscript has been modified accordingly.

**Figures**

Fig. 1: the color scale makes it difficult to distinguish whether the bed is below sea level or slightly positive. I would suggest adjusting the scale so that there is a clear transition between positive and negative base elevation. If the colorbar is modified in this figure, it should be modified in Figures A2-A4.

Thank you for the comment. We have adjusted the colorbar so that the transition between red and blue hues now aligns with the transition between bedrock above and below sea level.

**References**

Berdahl, M., Leguy, G., Lipscomb, W. H., Urban, N. M., & Hoffman, M. J. (2023). Exploring ice sheet model sensitivity to ocean thermal forcing and basal sliding using the Community Ice Sheet Model (CISM). The Cryosphere, 17(4), 1513-1543

Leguy, G. R., Lipscomb, W. H., and Asay-Davis, X. S.: Marine ice sheet experiments with the Community Ice Sheet Model, The Cryosphere, 15, 3229–3253, https://doi.org/10.5194/tc-15-3229-2021, 2021.

Gomez, N., Weber, M.E., Clark, P.U. *et al.* Antarctic ice dynamics amplified by Northern Hemisphere sea-level forcing. *Nature* **587**, 600–604 (2020). https://doi.org/10.1038/s41586-020-2916-2

---

## Author Comment (AC2)

**General comments**

In the present work, the authors study how the grounding line retreat of Thwaites Glacier, as predicted until 2350, is influenced by the spatial resolution of the ice sheet model, the GRD model, and by the temporal resolution used to couple them. In particular, they vary these resolutions independently from each other and observe significant differences of the forecasted GMSL contribution, which mostly depends on the ice sheet resolution for short prediction horizons (2100), and on a combination of the resolution parameters for longer horizons (2350). The observed differences are explained by the fact that high resolution runs capture bathymetric features more accurately, which can be enhanced by GRD effects. The latter is well exemplified by showing how the grounding line retreat varies if the isostatic adjustment is neglected, purely elastic or viscoelastic.

I believe that the manuscripts submitted by the authors is a valuable contribution to the literature, since it provides a more "orthogonal" analysis of the three main resolution parameters than what was made to date. I appreciate that many permutations of the model setup were made to highlight the main sources of difference in future GMSL contributions and I largely agree with the mechanistic explanations given in the paper, which are well supported by the figures. Overall, I enjoyed reading the article, which is well written, instructive and correctly embedded in the present research context.

Before a possible publication, I therefore only envision minor revisions, which are listed below. I wish all the best to the authors for the rest of the publication process and look forward to reading the final version of the manuscript.

Best regards,

Jan Swierczek-Jereczek

We would like to thank Jan Swierczek-Jereczek for his positive and thorough review of our manuscript. His comments, which we address in the sections below, have helped advance our investigation and brought valuable new insights to light. They have also significantly improved the clarity and visual presentation of our figures. In addition, we are grateful for his suggested reformulations, which have made several parts of the text more accessible to readers. Overall, we deeply appreciate his contribution, which has greatly enhanced the quality of the manuscript.

**Specific comments**

1.  The title is somewhat vague: you don't mention the exploration of various resolutions, which is the central contribution of the work. I believe something like "Spatiotemporal model resolution significantly affects the interaction between the solid Earth and Thwaites Glacier" is more coherent with your storyline. I understand that you want to mention ridges because they are the

explanation you find for this sensitivity… but "Insights from Reinforced Ridges" does not convey this adequately in my opinion.

Thanks for this comment. We have modified the manuscript's title to explicitly reference the 'resolution requirement' aspect of the paper. We have also rearranged the sentence's order to clarify the role of reinforced ridges in explaining the observed sensitivities.

2. Your motivation could benefit from a clearer thread, which I try to convey in the forthcoming sentences: *ISMIP6 models forecast that grounding line retreat in the Amundsen Sea Embayment (ASE) will lead to the greatest GMSL contribution over the coming centuries. In this region, the upper-mantle viscosity is particularly low and implies that the GRD response, which provides stabilising effects on grounding line retreat, is particularly fast. The stabilising effects of GRD can be enhanced by the presence of ridges and confinements, which have been identified in ASE but can only be represented by using high model resolutions*. This is your central motivation to study the resolution dependence and deserves to be highlighted in a clearer way. This is particularly the case for the abstract, which I invite you to change accordingly.

Thanks for this great suggestion. We've weaved in this reformulated motivation into the abstract's first couple of sentences.

3. L. 11: "extends buttressing" → "delays grounding line retreat".

The abstract has been modified accordingly.

4. L. 18: "In the fastest melting region of the ice sheet, Thwaites Glacier has displayed signs of early collapse (Joughin et al., 2014)." → "In Thwaites Glacier, the fastest flowing region of the ice sheet (Rignot et al., 2014), early signs of a collapse have been identified (Joughin et al. 2014, van den Akker et al. 2025), although this is contrasted by other studies (e.g. Hill et al. 2023)". I include all additional references at the end of this document.

Thanks for this comment and these references, the manuscript has been modified accordingly.

5. L. 24: I believe this is the good moment in the paper to mention that, in the context of Antarctica, the order of importance of GRD feedbacks is: first deformation, second gravity and last rotation, the latter being almost negligible due to the high latitude. I think this would help the (non-expert) reader. Alternatively, you can mention this in the discussion.

Thanks for this suggestion, we've modified the manuscript accordingly.

6. L. 30: It feels much more accurate to say that the mantle rheology controls t*he time scale* of GRD effects (particularly the deformational aspect, which might be nice to emphasise), not the "strength" of it. I suggest correcting this in other places too (see below).

We agree with the reviewer on this point and have modified the manuscript accordingly.

7. L. 49: "Ice sheet models have several major components, including mesh structure, basal melt and surface mass balance (SMB) parameterizations, which may affect the sensitivities of coupled simulations to (i), (ii), and (iii). To account for such effects, we report our results for 2 widely different coupled model setups labeled SLIM and PLUS. These have the same GRD model but different ice sheet models representative of the spectrum of complexity of modern ice sheet models." → "Ice sheet models are subject to a wide number of modelling choices, which can greatly affect the sensitivity of the coupled simulations. To check the robustness of our assessment of the sensitivity to resolution, we report our results for 2 widely different

coupled model setups, labeled SLIM and PLUS, which differ in the imposed surface mass balance, the basal melt law and the mesh type."

I believe this captures a bit better your motivation: you introduce SLIM and PLUS because you want to show that the importance of the resolution does not depend on other key modelling choices (SMB, BMB and mesh type). It is in fact what you observe, since SLIM and PLUS look qualitatively the same in Figure 5. I feel like the latter is a very strong message of your paper and think it could be stated more emphatically.

Thank you for this reformulation, we have weaved it into the manuscript.

8. L. 55: How is warming applied to the ocean and to the atmosphere in SLIM? Are they just uniform anomalies? Please specify how you derive that from SSP5-8.5!

Thanks for bringing this up. We clarify that only the PLUS configuration applies climate forcing from the SSP5-8.5 scenario via CESM2 outputs, both for ocean and surface mass balance (SMB). In contrast, the SLIM configuration uses a simplified approach: ocean-induced melt is parametrized solely as a linear function of the ice shelf base, without any temporal variation or explicit climate forcing. Similarly, SMB is held constant in time, based on the 1979–2010 average from the RACMO2.1 regional climate model. To make this distinction clearer, we have added a summary Table (Table 1) comparing the key differences between PLUS and SLIM.

9. L. 64-65: This feels like a particularly low value of the model time step… Did you observe significant changes when using higher values? This might be important to discuss since the article is all about spatiotemporal resolutions.

Thanks for this comment, it has been addressed in RC1's 4[th] comment of the line-by-line section and the manuscript has been modified accordingly. We acknowledge that this is a conservative choice as we did not observe significant changes when using higher values (1 to 2 months) during preliminary analysis at project inception.

10. L. 66-67: Are you optimising for each of the model setups individually? This question applies to all the permutations of the paper (SLIM / PLUS, uncoupled / elastic / viscoelastic, varying resolutions). This is important to mention somewhere in the manuscript. Also, could you include a figure in the appendix showing the error with respect to present day after the optimisation? That would be very helpful.

Thanks for bringing this up. The friction and rheology parameters are optimized separately for the SLIM and PLUS setups. For all permutations (e.g., uncoupled / elastic / viscoelastic, and varying resolutions) within a given setup, we use the same optimized friction and rheology fields. This is to permit an "*orthogonal*" sensitivity study, where only the parameter or resolution of interest is varied, and all others are held constant. We have added this information in Appendix A2, alongside a new figure showing the error to present-day observations following the optimization.

11. L. 68-69: What do you gain from imposing this GRACE extrapolation? Does it change a lot from just imposing present-day thickness? From the reader's perspective, it is unclear how this can contribute to a more accurate experimental setup. In my opinion this is quite superficial since, if Thwaites is really the first to collapse, it is unlikely to be affected by what will happen in neighbouring Glaciers.

Thanks for this comment. We refer here to the response given in RC1's 5[th] comment in the Line-by-Line section: The added value of using Grace data is to account for ocean loading due to farfield ice mass change (Gomez 2020), notably in the first few decades of the simulation. In the end, the reviewer is right that the effect might be small even then, but we wanted to make sure that any possible effect of present-day rising sea level on WAIS initial stability was accounted for. We've modified the manuscript to include this information.

12. L. 78: "The Extended Burgers Material (EBM) (Ivins et al., 2022) rheology is…" → "As described in the methods, the Extended Burgers Material (EBM, Ivins et al., 2022) rheology is…".

Thanks for this comment. Here we have modified "The Extended Burgers Material (EBM) (Ivins et al., 2022) rheology is…" → "The Extended Burgers Material (EBM, Ivins et al., 2022) rheology is…" We have left out the "As described in the methods" part as we are still in the Methods section.

13. L. 105: "by the transient EBM rheology described in the Methods section, we compare…" → "by the transient EBM rheology, we compare…" You actually don't describe the EBM rheology in the methods and should briefly do so, since it is important to understand the parameters of Table 1.

Thanks for this comment, we have added a thorough description of the EBM rheology in section 2.2.

14. L. 102: The subsection title is quite misleading since it suggests that there is a single baseline setup, although you have two of them. Please change this.

Thanks for pointing this out, we have modified the subsection title and parts of the subsection's 1st paragraph accordingly.

15. Figure 1: "+ PICOP" → "+ PICOP & CESM ocean". Using the CESM ocean fields here might actually be the modification that has the largest impact! For instance, does CESM represent the warm-water intrusion observed over the last decades in ASE? I suspect not since this is quite difficult, but that of course leads to a massive reduction of the grounding line retreat! This could drastically change the statement made at l. 268-273, which I think is largely unfounded: from what I understand, the linear melt law depends on the depth of the ice shelf base, which is largely unaffected by GRD. I believe the driver here is rather the CESM ocean forcing and, unless you prove the contrary, I would remove Appendix 7 and show a difference map of the ocean forcing instead.

Thank you for this comment. We have revised the manuscript accordingly. Figure 1 now uses the "PICOP + CESM ocean" label to clarify that the PLUS setup uses PICOP forced with CESM2 ocean. We also agree that the original statement made in lines 268–273 lacked sufficient support and hence have removed it, along with Appendix 7. You are correct in pointing out that CESM2 does not fully capture the warm-water intrusion observed over the past few decades in ASE. Furthermore, as you noted, the linear melt law used in SLIM depends only on the ice shelf base depth. This however means that there is no ocean forcing (temperature or salinity) in SLIM so we cannot make a map to show the difference in ocean forcing between SLIM and PLUS.

16. Figure 2: I am quite surprised that the comparatively small elastic bedrock uplift (max 8 m) has such a strong influence on SLIM (and SLIM + CESM2 SMB), as shown in Figure 1. The change in grounding line position between uncoupled and elastic run is massive, whereas it is barely noticeable as soon as you use PICOP. This might be related to the previous point….

Alternatively, it might be due to how you apply melt at the grounding line, which you really should mention somewhere here.

Thank you for your comment. In the revised manuscript, we have clarified how melt is applied at the grounding line, alongside our sub-element parameterization of grounding line migration. We agree that the impact of elastic bedrock uplift appears much more pronounced in Figure 1 once PICOP is introduced. However, direct comparisons between Figures 1b and 1c are not straightforward because the total ice mass loss (and hence the solid Earth feedback) differs substantially between these setups at the same point in time. If we instead consider grounding line positions at similar uncoupled grounding line configurations in Figures 1b and 1c, the negative feedback on ungrounded ice area looks much more comparable. Therefore, it is difficult to pinpoint the causes of this apparent discrepancy, and further investigation is necessary to fully resolve this issue.

17. L. 117: "As expected" → I wouldn't expect the SMB of CESM2 SSP5-8.5 to provoke a grounding line retreat, since it probably presents a substantial increase in precipitation, while maintaining surface temperatures that are largely below 0°C. I guess your answer to Point 8 might clarify this difference. If not, please provide an explanation.

Thank you for this comment. We agree that "as expected" was a poor choice of wording here and we have revised the manuscript accordingly. To clarify: although the CESM2 SSP5-8.5 SMB does show an increase in snow accumulation relative to the average RACMO2.1 climatology taken in SLIM, this increase is primarily concentrated in between ~2050 and 2200. Before 2050 and after 2200, the CESM2 SMB is smaller than the RACMO2.1 average in the region of interest –with higher negative values near the grounding line of Thwaites Glacier.

This is consistent with the behavior of the grounding line of both simulations: the retreat under CESM2 SMB forcing is similar to that of the RACMO-forced SLIM simulation up until ~2200, after which a more pronounced inland retreat occurs, matching the timing of SMB becoming more negative near the grounding line. This is now clarified in the revised manuscript text.

18. Figure 4 and the corresponding text explanations are particularly nice!
Thanks !

19. L. 160 and following: I understand that defining an error threshold is particularly convenient for the analysis since it allows a simple statement like "resolution sufficient" or "resolution insufficient" (and it also allows a comparison to Wan et al., 2022). However, I believe that such a statement depends on the application case and I would therefore recommend phrasing the rest of the results in a less binary way (which does not prevent a comparison to Wan et al., 2022). For instance, an error of 6% is totally acceptable in almost all cases, especially given that there are many additional sources of error (the laterally variability of Earth properties for instance). In particular, this would prevent sentences like "Overall, we find that the GRD model's response can be resolved at 111 km (n = 180) and still verify our 5% condition through 2350." (l. 184), which, as you know, is completely wrong for laterally varying viscosities. Please discuss this sentence if you want to keep it! Along this line, it should be highlighted that n=180 (at l. 280) is a lower bound on the Love number degree, since you have a 1D Earth.

Thanks for these great comments! We've added the following sentences to add nuance and discuss each of these statements so they wouldn't be misinterpreted by the reader:

ex-L160: "Although a 5% threshold is chosen for convenience in this study, we note that the practical acceptability of this error depends on the intended application and its importance relative to other sources of uncertainty."

ex-L184: "Importantly, we note that this (*the spatial resolution of the GRD response*) does not refer to the resolution at which the lateral variations in mantle rheology are resolved in 3D GRD models. [...] We find that the solid Earth's responsecan be resolved at 111 km (n=180) and still verify the 5% condition through 2350 in our GRD model"

ex-L280: "Further investigation is needed to determine whether GIA models accounting for lateral viscosity variation are typically more sensitive to higher mesh resolution in conditions comparable to our simulations. In particular, this sensitivity may depend on the resolution of the input seismic tomography model in the lithosphere and sublithospheric mantle structure."

20. L. 165 and following: you only mention differences in GMSL and ungrounded area, without specifying the sign of it (some exceptions as e.g. l. 174). This is of course conditioned by the fact that you don't include this information in Fig. 5 either. Including the sign in both the description and the figure would make everything much easier to read!

Thanks for this comment, we have updated both the manuscript and Fig. 5 accordingly.

21. L. 167 and following: you begin describing the results with a lot of detail. I feel like an overall picture prior to that would be welcome. Something like: "For SLIM as well as for PLUS, coarsening the resolutions invariantly produces a qualitatively similar reduction of accuracy. This effect is dominated by the ice sheet model resolution for short prediction horizons (2100), whereas the GRD resolution and the coupling time step become similarly important for longer horizons (2350). In particular…"

This for this suggestion ! We have modified the manuscript accordingly.

22. L. 192-199: You might want to cite Williams et al. (2025), which is the latest work related to this.

Thanks for pointing this out, we've included this reference in our manuscript.

23. Figure 5: It is clear why the error would increase over time but... can you explain why it decreases over time in the case of the ice sheet model? I guess the *absolute* error increases but, because the GMSL contribution increases faster, and this leads to a decrease of the *relative* error. Can you confirm? If yes, please mention this in the text.

Thanks for bringing this up. This is in fact due to a combination of factors. Indeed, the relative error decreases as the absolute values increase. Additionally however, in the case of PLUS, we observe that the error switches from positive to negative. In other words: the grounding line of the coarsen resolution goes from further inland to further to-sea compared to the baseline as time goes by. This entails small differences to the baseline close to the time the switch happens. We note that this behavior could be explained by coarser resolutions which lower stabilizing bedrock highs and elevate bedrock lows (weaknesses in the topography). We have included this information in the text (Section 3.3. and 4.1) and have updated figure 5 to provide information on the positive or negative signs of the difference to the baseline.

24. Figure 5: even the highest resolution used here has not converged since it displays significant differences in GMSL contribution with e.g. +1km for the ice sheet model resolution. This is unsurprising and completely ok but needs to be stated!

Manuscript modified accordingly

25. In almost all maps of Thwaites that you show, you have a white cut-out in the bottom left corner. This could easily be fixed and would make everything a bit prettier.

Unfortunately, due to the way the ASE domain is set up in our simulation, it was non-trivial to remove the cut-out. But we agree the white cut-out wasn't very pretty, so we have adapted the frames in a more aesthetic way.

26. The mask shown in Fig. A3 looks very arbitrary. I don't understand this choice, which does not look like it is motivated by a drainage basin. Please explain this.

Thanks for bringing this up. We agree that the mask in Fig. A3 may appear unorthodox compared to typical drainage basin boundaries. In the revised manuscript, we have clarified that the mask approximately corresponds to the Thwaites and Haynes glacier basins but is intentionally extended. This extension was designed to encompass the region where grounding line positions diverge significantly between the Viscoelastic and Uncoupled runs by 2350. We felt this approach was necessary to avoid underestimating GRD feedbacks on Ungrounded ice area and GMSL contribution. We have updated both the figure caption and accompanying text in Appendix A4 to better explain this choice.

**Technical corrections**

1.  You use "Amundsen Sea sector" a lot. I would suggest using Amundsen Sea Embayment (ASE), which is becoming a standard acronym.

Thanks, this lightens the text, we've modified the manuscript modified accordingly.

2.  L. 2: "century timescale" → "centennial timescale". Also at line 15.

Manuscript modified accordingly.

3.  L. 22-23: "and more recently centennial (...) timescales" → "and more recently centennial timescales (...)." Also, you should add Gomez et al. (2024) here, which, to my knowledge, is the latest article that supports this point (especially since you already have it in your refs).

Manuscript modified accordingly.

4.  L. 23-24: "In the vicinity of the grounding line –the region in which grounded ice becomes afloat, these interactions mainly comprise stabilizing negative feedbacks from sea-level fall and bedrock uplift" → "In the vicinity of the grounding line, the region where grounded ice becomes afloat, these interactions mainly comprise negative feedbacks on grounding line retreat. They consist in a gravitationally-driven reduction of the regional sea-surface height and a bedrock uplift due to the reduction of the surface load applied on the solid Earth."

Thanks for this suggestion, the manuscript has been modified accordingly.

5.  L. 25: "The negative feedbacks may be regionally enhanced by low mantle viscosity"

→ "In ASE, the deformational feedback is faster than the global average, due to upper-mantle viscosities that are particularly low." Here, I believe you should cite Ivins et al. (2023) (which you already have in your refs) rather than Coulon et al. (2021) since the former infers upper-mantle viscosities whereas the latter is a simulation study based on relaxation times.

Thanks for suggesting this, we've modified the manuscript accordingly.

6.  L. 28: "Glacial Isostatic Adjustment –GIA" → "Glacial Isostatic Adjustment – GIA"

Manuscript modified accordingly.

7.  L. 31: "the strength of GRD effect" → "the timescale of deformational effects". (Of course this has repercussions on gravity and rotation, which, however, are of second order). Also, this is a much better place to cite Coulon et al. (2021)!

Manuscript modified accordingly.

8.  L. 59: "sub ice shelf water" → "sub-shelf meltwater"

Manuscript modified accordingly.

9.  L. 68: "the model's mesh cover…" → "the model mesh covers…"

Manuscript modified accordingly.

10. L. 78: "(Adhikari et al 2016)" → "(Adhikari et al., 2016)"

Manuscript modified accordingly.

11. Table 1 and throughout the document: the multiplication between units is represented by "." (low dot) when it should be a "·" (middle dot).

Manuscript modified accordingly.

12. Table 1: as you mention in the discussion, the viscosity of the asthenosphere and the upper mantle are rather in the lower range of literature values. I think this should be mentioned earlier because it conditions the interpretation of all the results you present: the differences between elastic and viscoelastic basically represent an upper bound!

Thanks for this comment, we have adjusted section 2.2 accordingly.

13. Table 1: I would make two distinct columns for \tau_H and \tau_L.

Manuscript modified accordingly.

14. L. 97: "appendix" → "Appendix". Also at lines 115, 219.

Manuscript modified accordingly.

15. L. 102 and following: I understand you write "Viscoelastic", "Uncoupled" and "Elastic" with a capital letter because you consider them to be substantives. I would be in favor of just writing them as simple adjectives (viscoelastic, elastic and uncoupled runs). This feels lighter and equally well understandable.

Manuscript modified accordingly.

16. Figure 1: I think a different colorbar would greatly improve the visualisation of the bedrock elevation (by the way, why do you prefer to use "ice sheet base elevation" in your colorbar

label?). For instance, something like what isused in Garbe et al. (2020) could be nice (with adjusted range):

[Figure]

Bed topography (metres above sea level)

Thank you for this suggestion, as well as for referencing Garbe et al. (2020). We have taken this and a similar comment from RC1 into account and revised Figure 1 (and Figures A3–A5) accordingly. We did try the suggested colorbar (and the "balance" colorbar from the cmocean package) but ran into contrast issues that made the grounding lines difficult to see—despite adjusting the grounding line colors. Because the color coding for grounding lines remains consistent throughout (e.g., green for the Viscoelastic case), finding a color palette that maintained clarity across all figures proved challenging.

Ultimately, we decided to adjust our existing colorbar limits to [-2000, 2000], so that red hues indicate bedrock above sea level and blue hues indicate bedrock below sea level. Regarding our initial use of "ice sheet base elevation," we intended to emphasize that the dark blue portions represent ice-free ocean. However, as we feel it is clear from the context that these are indeed ocean areas, we have plotted "bed elevation" instead in the revised manuscript.

17. All figures: the effort of using a large font for legibility is well appreciated but you might have overdone it a bit. The difference to the font size of the preprint is quite large sometimes (assuming the figures will have a roughly similar size in the final version).

Thanks for pointing this out. We have reduced the font size by 5 points in Figures 1 and 2. And by 10 to 15 points in Figure 4 (which we felt had the largest increase in font size compared to the main body of the manuscript). We have also removed the bolded characters in Figure 5 which contributed to making the font larger. We have also adjusted the Appendix figure sizes and fonts to avoid overly large fonts.

18. L. 110: "depth dependent" → "depth-dependent". I think the dash is helpful here since you are concatenating many substantives.

Manuscript modified accordingly.

19. L. 113: "SP5-8.5" → "SSP5-8.5"

Manuscript modified accordingly.

20. L. 121: there is a weird indentation here. Also at lines 124, 164.

Thanks for pointing this out, this has been fixed.

21. Figure 3 is very convincing but would benefit from having separate panels for GMSL contribution, ungrounded area and GRD feedback (right now the y-axes are very confusing). I would suggest something like:

[Figure]

Figure 3 has been modified accordingly.

22. Figure 4 is also great albeit improvable. It would benefit from having a larger orange circle in panel (a), which corresponds better to the ridge. Since the ridges are spatial regions, red and orange vertical shadings in panels (b) and (c) are much more appropriate than the circles used in the current version. I would welcome a similar plotting style for panels (b) and (c) (right now they present different frame widths and line styles for the grid). "GL delay induced by GRD effects" → "Delay of GL retreat due to GRD effects". Please make the colorbar of panel (e) discrete instead of continuous in order to match the map. Finally, panel (d) could be deleted altogether since it does not provide useful information apart from the rectangle delimitation and the arrow, which could also be easily placed in panel (e). If you decide to keep panel (d), please remove the white mask beyond the GL retreat, which actually only applies to panel (e).

Thanks a lot for these suggestions and comments ! We have taken all of them into account and modified Figure 4 accordingly. Please note that following your suggestion, we have removed the circles altogether (even in (a)) and replaced them with shadings which we feel better communicates the spatial extent of the ridge. We have also opted to keep panel (d) as it provides a way to directly match the red isocontour of figure (e) on the bedrock map (and notably on the ridges). Hence, as suggested, we have removed the white mask to show the bedrock beyond the GL retreat.

23. L. 148-149: remove "The red line (Figs. 4d, e) represents the 25 year delay isocontour which coincides with the location of the second ridge (Fig. 4d)." You already mentioned that in the figure caption.

Manuscript modified accordingly.

24. L. 162: "degrading" → "varying"

Manuscript modified accordingly.

25. L. 180: "Love number degree 'n' which is the…" → "Love number degree, $n$, which is the…".

Manuscript modified accordingly.

26. L. 226: "that in this study, the" →"that, in this study, the"

Manuscript modified accordingly.

27. L. 236: "Nevertheless, this study provides a reproducible framework for analyzing the individual influences of various modeling choices." I think you should rephrase this since testing parameters independently from each other is nothing conceptually new. In my opinion, you are not providing a framework but just complying with good scientific practice to isolate the individual parameter sensitivities.

Manuscript modified accordingly.

28. L. 245: "based on the ISSM" → "based on ISSM". More coherent with how you treat acronyms like SLIM, PLUS, PICOP....

Manuscript modified accordingly.

29. L. 251: "An aggressive yet realistic solid Earth parametrization is deliberately" → "A fast yet realistic parametrization of the viscoelastic uplift is deliberately"

Manuscript modified accordingly.

30. L. 252: quotation marks look weird, please correct.

Manuscript modified accordingly.

31. L. 264: (Swierczek-Jereczek et al., 2023) → (Swierczek-Jereczek et al., 2024). You are citing the preprint although there is a published version.

Manuscript modified accordingly.

32. L. 265: "on computationally intensive complex ice sheet dynamics" → "on computationally expensive ice sheet dynamics"

Manuscript modified accordingly.

33. L. 313: "Volume Above Floatation (HAF)" → "Height Above Floatation (HAF)". I am not sure this section really brings useful information compared to simply referring to Adikhari et al. (2020).

Thanks for catching this typo. We have also removed the brief overview part and simply refer to Adhikari et al. 2020.

34. In the references, some acronyms are written oddly: e.g. "sE", "gMD", "tC"...

Thanks for catching this !

**Additional references**

1. van den Akker, T., Lipscomb, W. H., Leguy, G. R., Bernales, J., Berends, C. J., van de Berg, W. J., and van de Wal, R. S. W.: Present-day mass loss rates are a precursor for West Antarctic Ice Sheet collapse, The Cryosphere, 19, 283–301, https://doi.org/10.5194/tc-19-283-2025, 2025.

2.  Hill, E. A., Urruty, B., Reese, R., Garbe, J., Gagliardini, O., Durand, G., Gillet-Chaulet, F., Gudmundsson, G. H., Winkelmann, R., Chekki, M., Chandler, D., and Langebroek, P. M.: The stability of present-day Antarcticgrounding lines – Part 1: No indication of marine ice sheet instability in the current geometry, The Cryosphere, 17, 3739–3759, https://doi.org/10.5194/tc-17-3739-2023, 2023.

3.  Williams, C.R., Thodoroff, P., Arthern, R.J. *et al.* Calculations of extreme sea level rise scenarios are strongly dependent on ice sheet model resolution. *Commun Earth Environ* **6**, 60 (2025). https://doi.org/10.1038/s43247-025-02010-z

4.  Swierczek-Jereczek, J., Montoya, M., Latychev, K., Robinson, A., Alvarez-Solas, J., and Mitrovica, J.: FastIsostasy v1.0 – a regional, accelerated 2D glacial isostatic adjustment (GIA) model accounting for the lateral variability of the solid Earth, Geosci. Model Dev., 17, 5263–5290, https://doi.org/10.5194/gmd-17-5263-2024, 2024.

5. Gomez, N., Weber, M.E., Clark, P.U. *et al.* Antarctic ice dynamics amplified by Northern Hemisphere sea-level forcing. *Nature* **587**, 600–604 (2020). https://doi.org/10.1038/s41586-020-2916-2